# Oncologists' views on ototoxicity monitoring in head and neck cancer patients: A South Indian qualitative study

Varsha Shankar[1], Jayashree Seethapathy[1]*, Satish Srinivas[2], Raghu Nandhan[3], Prasanna Kumar Saravanam[4]

1 Department of Audiology, Sri Ramachandra Faculty of Audiology and Speech Language Pathology, Sri Ramachandra Institute of Higher Education and Research (Deemed to be University), Chennai, Tamil Nadu, India, 2 Department of Radiation Oncology, Sri Ramachandra Institute of Higher Education and Research (Deemed to be University), Chennai, Tamil Nadu, India, 3 Department of ENT – Head & Neck Surgery, Madras ENT Research Foundation, Chennai, Tamil Nadu, India, 4 Department of ENT, Head and Neck Surgery, Sri Ramachandra Institute of Higher Education and Research (Deemed to be University), Chennai, Tamil Nadu, India

☯ These authors contributed equally to this work.
* jayashree.s@sriramachandra.edu.in

**Data Availability Statement:** All relevant data is within the manuscript and its Supporting information files.

## Abstract

### Aim

The perspectives and practices of healthcare professionals regarding ototoxicity in individuals with head and neck cancers are important for the implementation of ototoxicity monitoring. The current study aims to explore the oncologist's awareness and perspectives of ototoxicity and ototoxicity monitoring for individuals with head and neck cancer in a South-Indian district, using qualitative semi-structured interviews.

### Method

The COnsolidated criteria for REporting Qualitative research (COREQ) Checklist was used to guide the method of the current qualitative study. A conceptual framework was developed for the formulation of the interview guides. Three medical oncologists and six radiation oncologists from cancer care centres participated in the study. The interviews were audio recorded and transcribed verbatim. Thematic analysis was carried out using a hybrid inductive-deductive approach to present the findings under the respective overarching themes.

### Results

All oncologists are aware of the ototoxic effects caused by radiation therapy and chemoradiotherapy. It was consistently reported that the severity of the hearing loss was not substantial enough to warrant significant concern. Ototoxicity is not emphasized during the counseling process. All participants reported having awareness and knowledge of ototoxicity monitoring programs and understood their importance. However, none of them reported the implementation of an ototoxicity monitoring program in their facility.

**Funding:** The author(s) received no specific funding for this work.

**Competing interests:** The authors have declared that no competing interests exist.

## Discussion

It is imperative to enhance the knowledge and understanding of the ototoxic nature of cancer treatment modalities among oncologists. It is crucial to raise awareness regarding the significance of ototoxicity monitoring programs among all physicians involved in the treatment of patients with cancer for maximum impact. The barriers to implementing ototoxicity monitoring programs in high- and low-income countries are similar. The models implemented in high-income countries can be adapted for use in low-middle income countries with suitable restructuring.

## Conclusion

The current study provides valuable insights into the status of ototoxicity monitoring in the South-Indian context. The findings align with the key components of the health belief model, including perceived susceptibility, perceived barriers, and cues to action. Involving oncologists in the planning phase of the ototoxicity monitoring programs can help tailor future research questions and solutions to improve quality of life, foster collaboration among healthcare professionals, and produce actionable outcomes that may influence policy on ototoxicity monitoring.

## Introduction

In India, head and neck cancers (HNC) are one of the most common malignancies reported in both males and females [1]. One in 33 males and one in 107 females are at risk of developing the disease in their lifetime [2]. Treatment for HNCs has evolved over the past decade, and today multi-modality therapy consisting of radiation therapy (RT), along with surgery, chemotherapy (CT), and immunotherapy, cures an increasing percentage of patients with locally advanced cancers. Conservative surgery or RT is equally effective in early stages of HNC (I and II). Locally advanced stages III and IV require radical resection, reconstruction, and post-operative RT or concurrent chemoradiotherapy (CCRT) [1]. Patients with metastatic cancer are palliated with a combination of short-course RT and systemic therapy for symptom relief and to maintain or improve quality of life [3].

Surgical oncologists (SO), medical oncologists (MO), and radiation oncologists (RO) collaborate to provide treatment for patients with HNC. The treatment depends upon the stage, site, and complexity of the disease [4]. While post-treatment survival rates and life expectancy have increased, these treatment options may have a negative impact on the patient's quality of life due to the effects on swallowing, speech, hearing, and overall psychological well-being [5]. Thus, the treatment process also involves an integrated multidisciplinary care approach with periodic referrals to dentists, clinical psychologists, and speech and swallow therapists to manage side effects such as mucositis, odynophagia, and dysphagia [6]. These referrals are periodic and streamlined, due to the impact of these side effects on the prognosis of the disease [6].

It has been documented that RT and CRT can cause ototoxicity by impacting the structures of the inner ear [7]. Ototoxicity is the cellular degeneration of cochlear and/or vestibular tissues that leads to functional deterioration due to the use of certain therapeutic agents [8]. The incidence of sensorineural hearing loss (SNHL) after RT and CRT ranges from 0% to 43% and 17% to 88%, respectively [9]. In addition to the existing burden of disease, hearing loss can

lead to communication difficulties and social disengagement. It can lead to challenges in cognition, child development, and academic achievement among children. In adults, RT/CRT can result in poor balance, which interferes with walking and increases the risk of falling. Ototoxicity is also linked to depression and anxiety [10]. Thus, hearing loss due to RT/CRT is an important "quality-of-life" issue that should not be overlooked.

Ototoxicity monitoring (OM) tracks hearing and vestibular changes over time and alerts the consulting physician when changes are detected early. This can result in opting for alternative treatment protocols, possibly with less ototoxic medications [11]. There are several OM guidelines for patients receiving RT or CRT [12, 13]. According to ASHA and AAA guidelines, OM needs to be carried out at baseline, during treatment, and at 1 month, 3 months, and 6 months post-treatment [12, 13]. A coordinated referral from oncologists to the audiologist is essential to optimizing OM. A few studies explored the perspectives and practices of oncologists in South Africa and the United States of America (USA) regarding referrals to audiology for OM.

The perspectives and practices of healthcare professionals regarding cisplatin-associated ototoxicity in individuals with cancer were described in a hospital-based study from South Africa [14]. The study findings indicated that OM was not included in the CT protocols and no ototoxicity monitoring program (OMP) was implemented. All audiologists reported that the oncology department does not refer patients for routine OM. Similarly, another study used a written survey/survey-based interview to report the perspectives of four physicians on OMP provision in the United States (US) [15]. The interviewees reported that physicians' approaches to OM varied, with only some ensuring routine referrals to audiology while others relied on patient-reported symptoms and self-referrals. The referral processes were based on the respective oncologist's perceptions of the expected risk of ototoxicity.

In recent years, efforts in India have increased to establish regional cancer centres in rural areas and upgrade medical institutions with oncology departments to address the population's cancer care needs. It is worth noting that there are several tertiary cancer centers with state-of-the-art diagnostic workup and treatment protocols that improve the detection and treatment of various types of cancers [16]. However, these protocols are yet to meet international standards; the need of the hour is to focus on improving the quality of life of cancer patients.

Despite the prevalence of HNC in the Indian population, there is limited evidence documenting the post-treatment quality of life in Indian literature [17]. There is a lacuna pertaining to literature on periodic OM in India [14], and ototoxicity research is still in its early stages [11]. There are no studies reporting the practices and awareness of ototoxicity among oncologists treating cancer patients in the South-Indian context.

Thus, the current study aims to explore the oncologist's awareness and perspectives of ototoxicity and OM for individuals with HNC in a South-Indian district, using qualitative semi-structured interviews (SSIs). The specific objectives of the study are (a) to understand the current practices of oncologists in the treatment of individuals with HNC, and (b) to understand the oncologist's perspectives regarding ototoxicity and OMPs for individuals with HNC undergoing RT and CRT.

## Materials and methods

The study used a qualitative cross-sectional snapshot study design to accurately understand the current status of oncologist's awareness and perspectives of ototoxicity and OM [18]. The study utilized the COnsolidated criteria for REporting Qualitative research (COREQ) checklist [19] to guide the method (S1 Appendix).

### Research team and reflexivity

**Personal characteristics.** Two female authors (VS and JS) conducted the SSIs. Both authors were trained in qualitative research and have prior experience with conducting SSIs. The first author (VS) is a full-time Ph.D. research scholar, and the second author (JS) is an associate professor and head of the audiology department, having completed her Ph.D.

**Relationship with participants.** No relationship was established prior to the commencement of the study, and participant knowledge of the interviewers was minimal. The interviewer's interests in the topic and the reason for conducting the SSIs were clarified to the participant prior to obtaining consent from all the participants.

### Study design

**Theoretical framework.** A conceptual framework (Fig 1) was developed based on the constructs of the Health Belief Model [20] and a review of the literature. It was utilized for the formulation of the interview guide. The themes have been operationally defined within the context of the current study (Table 1). The researchers used content analysis [21] as the methodological orientation for the current study.

**Participant selection.** Medical oncologists (MOs) and ROs from cancer care centers in Chennai, Tamil Nadu, India, with a large number of patients with HNCs, were approached to participate in the study. Only MOs and ROs meeting specific inclusion criteria were invited to participate in the study; participants were expected to have at least one year of experience in the management of HNC. They had to be practicing oncologists in the district of Chennai. Oncologists with less clinical experience in the field of HNC and surgical oncologists were excluded from the study. In addition, oncologists who were involved in the ideation of the current study were excluded [22].

The researchers obtained the contact numbers of several oncologists through known contacts and colleagues in the field of medical and radiation oncology. In addition, some of the participants referred the interviewers to other MOs and ROs; thus, the participants were selected using a snowball sampling method [23]. Participants were approached through phone calls to describe the study, obtain consent, schedule an appointment, and conduct the SSI. Overall, approximately 20 MOs and ROs from these centers were approached in-person or through known contacts for participation in the study. However, a few MOs (n = 7) and ROs (n = 4) were unwilling to participate in the study; the main reasons included unavailability (n = 4), lack of time to participate (n = 3), lack of interest in the topic (n = 2) and not being aware of the topic to share insights (n = 2).

**Setting of data collection.** The interviews were conducted at the cancer care centers. With the exception of the researchers and participants, no additional individuals were present during the interviews to ensure confidentiality.

**Data collection.** The interview guide (Figs 2 and 3, S2 Appendix) was pilot tested on two participants and was modified based on the participant's feedback. Verbal informed consent was obtained before the interviews, and it was audio-recorded. The interviewers used questions and prompts from the interview guide to conduct the SSIs, which lasted approximately 30–45 minutes. Repeat interviews were not carried out. The interviews were audio recorded, and field notes were maintained during the interviews. Later, the audio recordings of the interviews were transcribed verbatim, and the transcripts were verified by the participants. As soon as redundancy was detected in the acquired data, data saturation was attained [24].

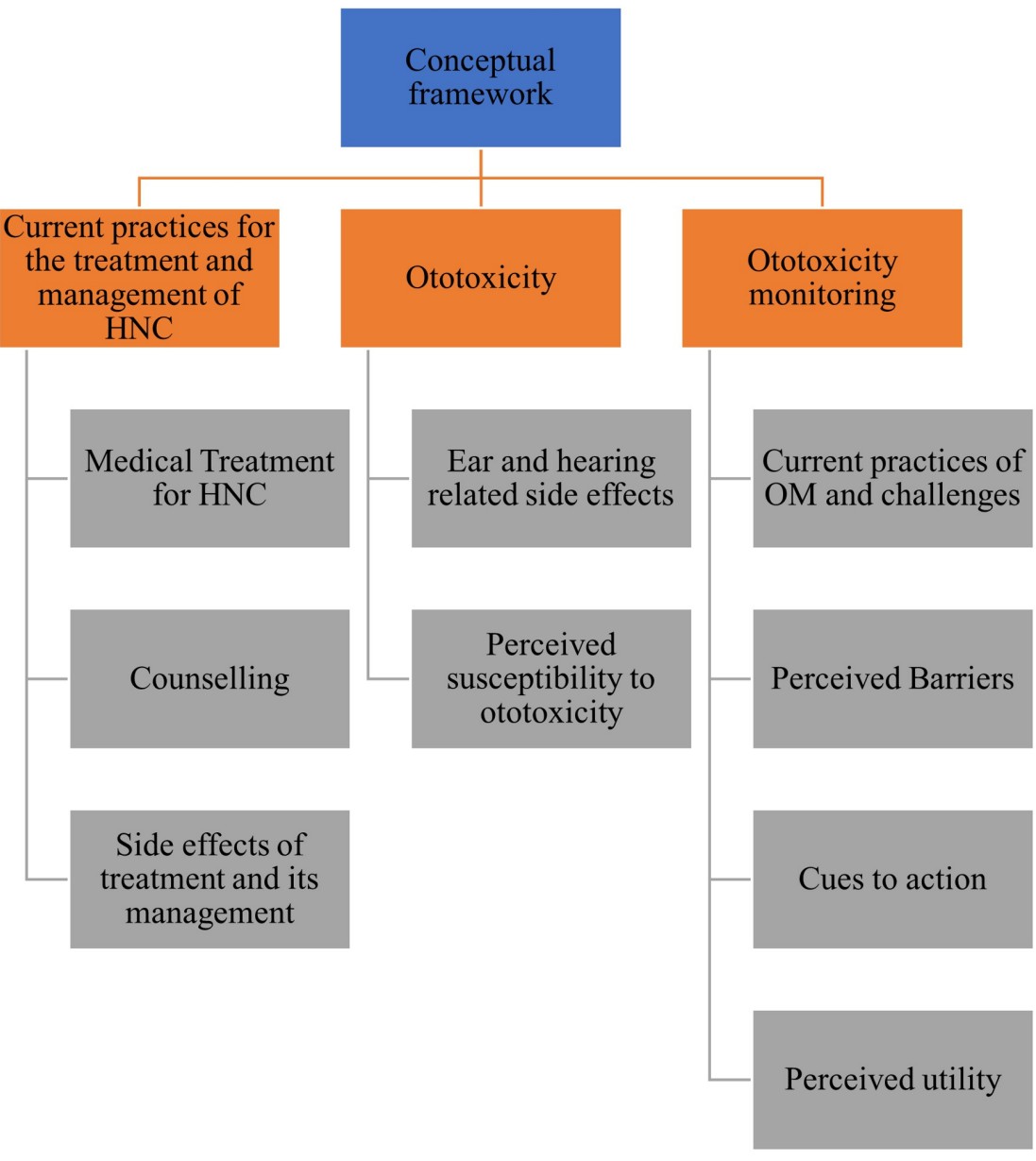

**Fig 1. Theoretical framework developed and used for the current study.**

## Analysis and findings

**Data analysis.** Thematic analysis was carried out using a hybrid inductive-deductive approach [25] with the Qualitative Data Analysis (QDA) Miner Lite software [26], a free software that allows the researcher to code and analyze qualitative data. Thematic analysis, as recommended by Braun and Clarke [27, 28], was carried out as follows:

*Familiarization with the data*: The authors became familiarized with the transcripts and added new codes to the current codebook based on inferences drawn from the interviews.

*Coding*: Two authors (JS and VS) independently coded the data and provided a description of the coding tree. Coding inconsistencies and differences were discussed and rectified. The

**Table 1. Operational definition of themes used in the theoretical framework.**

| S. No | Theme | Definition |
|---|---|---|
| 1. | Current Practices | Current practices used by Oncologists in the treatment of individuals with Head and Neck Cancer–treatment options, techniques used, counselling, side effects, management of side effects, audiological monitoring etc. |
| 2. | Perceived susceptibility | Oncologist's belief about an individual with HNC at risk of developing ototoxicity |
| 3. | Perceived barriers | Oncologist's belief about the challenges in implementing an OMP |
| 4. | Cues to action | Oncologist's suggestions and strategies for the future implementation of OMPs |
| 5. | Perceived Utility | Oncologist's perceptions regarding confidence to take action, perceptions about the potential success and impact of the OMP |

**Interview Guides**

Interview Date: _________
Interview time: _________

**Hello Dr.….**
I am Varsha Shankar, an Audiologist and Speech Language Pathologist working as a PhD Scholar at the Faculty of Audiology and Speech Language Pathology at SRIHER (DU).

I thank you for giving consent to participate in the study.

This interview is designed to understand the current practices, knowledge and perceptions of audiologists and oncologists about monitoring ototoxic effects in patients with head and neck cancer who are receiving CRT/ RT. The interview will last approximately 45 minutes to an hour. Before we begin, please let me know if you have any questions that I can answer or clarify.
The interview will now begin. I'm going to start audio recording our conversation for transcription and data analysis. The information will be kept strictly confidential, and your identity will never be revealed.

**Interview begins**

Dear Dr.__________
Guides: (Probes have been italicized; constructs have been highlighted in yellow)
1. How many years have you been practicing?
2. Describe your client base.
3. Please describe the current protocol followed at your hospital for the treatment of patients with head and neck cancer? *(Current practices)*
   *(Chemotherapy – dosage, types of drugs, Radiation therapy – techniques used - IMRT, IGRT)*

**Only For Oncologists:**
4. How do you counsel patients regarding the treatment and its side effects?
5. *(Counselling regarding hearing loss, changes in balance and tinnitus especially before beginning the treatment)* *(Current practices)*
6. Please describe some **common complaints** of patients with head and neck cancer undergoing RT/CRT? How do you deal with them? *(Perceived susceptibility)*
   (OR)
   What are the possible side effects of these treatments?
   When do you typically notice these symptoms and what type of medication were they on?
7. How important/serious are these complaints? *(Perceived severity)*
   *(How would the findings of these side effects influence treatment decisions?)*
   How do you currently manage these complaints?

**Oncologists and Audiologists:**
8. What are ototoxicity monitoring programs (OMP)?
9. What is your opinion about OMP? *(Benefits of OMP)* *(Perceived benefits)*

**Fig 2. Interview guide.**

10. Could you briefly describe the current status of OMP at your institution? *(Current practices)*
*(On-going/ started and discontinued/ non-viable/ waiting for it to start as routine?)*

11. How does the referral process work in your institution? *(Current practices)*
*(Do you refer all patients undergoing RT/CRT for audio-vestibular evaluation?)*
*(Do you refer patients selectively? If so, why? What in your opinion are the predictive factors used to refer patients undergoing RT/CRT for audio-vestibular evaluation?)*
*(Do you refer these patients for any other screening procedures?)*

12. Are you aware of any other such existing programs in Chennai? *(Current practices)*
*(If yes – probe for details, if no – probe for interviewee's perception about why clinics have not implemented OMPs)*

13. What do you think are the key elements of an OMP? *(Perceived barriers)*
(OR)
Could you describe any resources that can influence (either positively or negatively) the implementation of an OMP?
- *Financial/Economic*
- *Human - professional/ care takers/ technical help/ facilitator*
- *Infrastructure, Equipment/ technology, Technical/internet*
- *Organizational factors – acceptance, policy making/decision, SOP, Provider comfort/ ease/ acceptance/ perceptions?*
- *Patient's comfort/ Acceptance/ Perceptions/ Sociocultural aspects/ Inherent patient related factors with respect to ear and hearing care?*
- *Motivators/ influencers*
- *Evidence base for implementation of OMPs in the current context*
- *Any non-adaptable factors?*

14. Would you modify your current clinical practices to include OM? Why or why not? *(Cues to action)*

15. If you plan on implementing an OMP, how confident are you regarding it's sustainability? What are the challenges or facilitators you would expect? *(Perceived barriers and Self-Efficacy)*

16. In your opinion, who are the role players in an OMP?
*How should the referral pathway be structured for an effective OMP?*
*Whose responsibility is it to provide patients with information about the possible ototoxic effects of medication?*
*What do you believe is the appropriate schedule of hearing monitoring tests for the results to be useful?*

Is there anything else you would like to say regarding ototoxicity and OM?

**Interview ends**

Thank you for your participation in the study. We will have the data analysed and will keep you updated regarding the progress of the study.

**Fig 3. Interview guide.**

remaining transcripts were coded using the refined codebook, and new codes were assigned to unique and significant information discovered during this process.

*Searching for themes*: The codes were organized according to similarities and the frequency of occurrences, which assisted in classification. These categories were analyzed to identify significant patterns in the data, and the findings were grouped under the respective overarching themes.

*Reviewing themes*: This was followed by an assessment of the themes with regard to their applicability and credibility. Certain themes were combined during this process in order to present the findings in a more streamlined manner.

*Defining and naming themes*: Each theme was analyzed and defined according to the information it conveyed. The themes and sub-themes were named accordingly.

**Reporting.**   Participant quotations have been presented to substantiate the themes and findings. The quotes have been identified using specific identifiers. Both major and minor themes are described, along with emphasis on unique findings.

## Ethical considerations

This qualitative study was approved by the Sri Ramachandra Institutional Ethics Committee (Reference No. IEC-NI/22/JUL/83/81) to ensure that all ethical considerations have been addressed. The current study adhered to meet all ethical considerations, and there were no conflicts of interest. Written and verbal consent was obtained from all participants. The interview participants were aware of the research study and agreed to participate voluntarily without any coercion. They understood the nature of the research and were informed about their right to withdraw at any time. The participants' personal information and data was stored securely. Data was anonymized to protect participants' identities. All interviews were considered sensitive, and the quotes and transcripts were not disclosed without explicit permission. The findings of the study were shared to the respective participant for feedback and "no objections for publication".

## Measures to ensure trustworthiness

The study addresses all four criteria for assessing trustworthiness [29, 30]: credibility, transferability, dependability, and confirmability.

**Credibility**: The accuracy and truthfulness of the findings were ensured through prolonged engagement, persistent observation, and member checks. The interviewers used questions and prompts from the interview guide to conduct the SSIs, which lasted approximately 30–45 minutes. This resulted in persistent observation and intensive contact, enabling the identification of patterns and unique themes during the data analysis phase. The continuous probes allowed for informal testing of the information provided by the participants.

**Transferability**: The findings of the current study might be applicable to other Low-Middle income countries' (LMICs). The study's context and findings are described with thick descriptive data to allow for comparison.

**Dependability**: The study's consistency and reliability were verified by the other three co-authors. In addition, the research design was planned meticulously from the ideation stage.

**Confirmability**: The study's findings are shaped by the participants' responses rather than researcher bias through the process of reflexivity.

## Results

Nine participants provided consent to participate in the interviews. The participant characteristics such as profession, years of experience, gender and participant setting are described in Table 2.

The results of the current study are explained and presented under the following themes identified in the conceptual framework (Table 3):

### Theme 1—Current practices for the treatment and management of HNC

**Medical treatment for HNC.**   All interviewees stated that the treatment options for HNC are determined by the type, stage, and other special considerations. The tumor staging is commonly categorized as early stage, locally advanced stage, and metastatic or recurrent stage. Early-stage tumors are treated with a single modality, which is either surgery or RT. Surgery

**Table 2. Participant characteristics—profession, years of experience, gender, setting.**

| S.No | Profession | Code | Years of experience | Gender | Setting |
|---|---|---|---|---|---|
| 1. | Radiation Oncologist | RO1 | 2.5 years | Male | Private quaternary care centre |
| 2. | Medical Oncologist | MO1 | 30 years | Male | Private quaternary care centre |
| 3. | Radiation Oncologist | RO2 | 25 years | Male | Public charitable comprehensive cancer centre |
| 4. | Radiation Oncologist | RO3 | 1.5 years | Female | Private quaternary care centre |
| 5. | Medical Oncologist | MO2 | 3 years | Female | Public charitable comprehensive cancer centre |
| 6. | Medical Oncologist | MO3 | 3 years | Male | Private quaternary care centre |
| 7. | Radiation Oncologist | RO4 | 8 years | Male | Private quaternary care centre |
| 8. | Radiation Oncologist | RO5 | 23 years | Male | Private multi-speciality quaternary care hospital |
| 9. | Radiation Oncologist | RO6 | 11 years | Male | Private multi-speciality quaternary care hospital |

and/or CCRT are used to treat locally advanced malignancies. The metastatic cancers are treated using CT alone.

> *"We can broadly classify it (HNC stages) as early stage, locally advanced stage and metastatic state or recurrent setting. So early stage, mainly the modality is of a single type, most of the times; either they go for surgery or they go for RT. . .The locally advanced stage, where the tumour is unresectable, we go for CRT. . ..The third setting is a metastatic setting. We only go with the CT because it's already stage four, where the cancer has spread everywhere. So, we will give a systemic treatment."*
>
> *–MO2*

Intensity Modulated Radio-Therapy (IMRT) was reported to be the most often utilized treatment plan for HNC.

> *"The technique used here is IMRT—intensity modulated radiotherapy, and along with concurrent chemo. . . So that's basically what we do most of the times."*
>
> *–RO4*

According to one interviewee, CCRT is used as a treatment modality as CT acts as a radio-sensitizer for enhancing the effects of RT.

**Table 3. Themes and sub-themes identified according to the theoretical framework.**

| Themes | Sub-themes |
|---|---|
| Current practices for the treatment and management of HNC | Medical Treatment for HNC |
| | Counselling |
| | Side effects of treatment and its management |
| Ototoxicity | Ear and hearing related side effects |
| | Perceived susceptibility to ototoxicity |
| Ototoxicity Monitoring | Current practices of OM and challenges |
| | Perceived barriers |
| | Cues to action |
| | Perceived efficacy |

*"RT is combined with the concurrent CT because it (CT) acts as a radio sensitizer to increase effects of radiation in HNC."*

*–RO1*

Cisplatin is the most often used platinum agent to treat HNC, according to all participants, and it is typically administered in two regimens: 100 mg/m$^2$ once in three weeks or 40 mg/m$^2$ every week for five to six weeks. The treatment regimen is chosen based on several factors, such as the patient's condition, co-morbidities, and toxicity profile.

*"We can give cisplatin weekly or three-weekly. If we are giving for three weeks, we give hundred mg per meter square. . . the creatinine clearance should be more than 60. If we have a doubt that this patient is susceptible to some kind of toxicities and we don't want to expose them to this or we just want to ensure a lesser toxicity profile, then we'll try to give him weekly cisplatin and see. So, for weekly cisplatin, the creatin clearance should be more than 50."*

*- MO2*

With the advent of modern linear accelerators, many RT techniques such as IMRT, Image Guided Radio-Therapy (IGRT), Volumetric Modulated Arc Therapy (VMAT) are available. In terms of imaging protocol, IGRT differs from IMRT in that pre-treatment image verification occurs with inbuilt on-board kilovolt (kV) cone beam CT imaging as opposed to the megavolt (MV) X-ray imaging used in IMRT. Because it avoids/limits dosage to all "organs at risk," IMRT is the treatment technique of choice for patients with HNC. As a result, it has fewer side effects and toxicities.

*"IMRT is the definitive modality. For head and neck, we don't use any other modality. There are other options out there—IGRTs and VMAT, but the IGRT only varies in an imaging protocol point of view. It does not vary in terms of technique. Technique wise, it still remains the same, it's only IMRT. Image verifications is done probably twice a week in IMRT, whereas in IGRT, we do it four days in a week."*

*–RO4*

An MO and RO reported that in certain conditions, neo-adjuvant treatment is administered prior to surgical removal of the tumor. This is typically performed in situations when specific attention is needed, such as when the preservation of the larynx is a priority or when there is trismus.

*"When they (the radiation oncologist) feel that the patient has got Trismus. After you give radiation, the Trismus can worsen. So that will become very difficult for patients. The quality of life will be affected. So, there's some settings where if we feel that the trismus is going to worsen the patient's condition, we will try to give initial neo-adjuvant CT, then the patient will undergo surgery. So that will also be individualized. Like we don't give neo-adjuvant CT for all patients. So, depending on the clinical situations, we plan and give neo-adjuvant CT for very small percent of patients."*

*–MO2*

*"If it's gonna be, let's say somewhere like a hypopharynx or a larynx where you are looking more in terms of laryngeal preservation, you give probably the definitive CTRT first and then*

*reserve surgical options as a salvage. So, it varies based on the site in the head and neck and also based on the stage."*

*–RO4*

The treatment technique is determined by the tumor characteristics (site, stage and volume of the disease) and the patient characteristics (age and general condition of the patient). According to one MO, the risk of toxicity in elderly patients is assessed using a scoring system which aids the physician in the drug selection process.

*"For patients more than 65 years of age, we have a scoring system to see what will be the risk of toxicity that the patient can have because of any CT drug. . .. there are some 10–12 factors to assess for the toxicity in elderly patients. So that also we take into consideration. If you feel that the toxicity score is high, then we also try and tell them that this is the amount of risk you're going to have. . ... If no risks are there, only then we go for cisplatin. Otherwise, then we will avoid cisplatin. Explaining the risk, we will give carboplatin or the other agents. If they're able to afford it, we give a targeted treatment with cetuximab."*

*–MO2*

In some instances, the choice of technique may be influenced by the patient's affordability.

*"Actually, ours is a non-commercial, non-profit making institution. While most other institutions are preferring either VMAT or IMRT, we leave the choice to the patient.. if they are able to afford it, we do offer them VMAT, which is the best technique by far. . ..The less affordable patients would go in for IMRT. And the average patients would go in definitely for a 3DCRT. And in case, totally free patients—we treat 40% of patients totally free also. So, for such patients we still go by the traditional 2D technique also."*

*–RO2*

One MO specified that treatment is decided in multidisciplinary tumour board discussions.

*"There's a protocol: From day one, once they're diagnosed with CA of a specific sub-site, after staging it we have a tumor board here. So based on the tumor board decision, we take up the cases. . .. The decision (for treatment) will be taken in the multidisciplinary board discussions that will be held. So, there we discuss and we plan for chemoradiation."*

*–MO2*

**Counselling.** Both MOs and ROs agreed that counseling is an essential component of the treatment process and is contingent on several factors, such as the site, stage, and prognosis of the disease. Treatment options, duration, and other relevant aspects are highlighted during the counseling session. All participants reported that counseling begins upon diagnosis of the disease and continues periodically during the course of treatment, accompanied by a timely review of the patient's status. This reportedly ensured that all patient queries were addressed periodically.

*"We counsel them regarding the prognosis of the disease, regarding the treatment duration, treatment, morbidity, and, even prior to starting treatment. And of course, on a day-to-day*

*basis, we have one of the consultants who will be in the department, while the others may be in the ward or in the OPD and there's a unit doctor who specifically assigned to all the treatment units. So, any day-to-day doubts that they may have will also be clarified by the unit doctor. And weekly once, a consultant would also review all the ongoing cases. So, the counselling starts from day one and continues throughout the course of treatment because different people have different doubts which they may not be able to express on day one."*

*–RO2*

In addition, one MO emphasized the importance of counseling patients regarding stopping the use of tobacco before treatment is initiated.

*"Most of the epidemic cancers in this country is due to tobacco. Either by chewing or smoking. At the outset, that we have to stop. The counselling should be done for the tobacco cessation. Either in the form of chewing or smoking. That should go parallelly. In fact, much prior to the start of the treatment, this aspect has to be counselled."*

*–MO1*

One RO reported that the counseling process in India differs from that of western countries. The patient is counseled only briefly about the disease and treatment. However, it is the caregivers of the patients who are given a detailed explanation about the prognosis and severity of the disease.

*"Regarding the outcome, the attenders will probably be explained. Because in the Indian setting, still that kind of an openness has not yet been established; because the attender strives to kind of hide facts from the patient. But we make sure that the patient knows that he has a malignancy, probably tell them that it is a curable treatment. But the quantum of disease may not be conveyed to the patient. But in the (United) States or in the other countries abroad, it's the other way around. The patient gets to know first and only then the attenders get to know."*

*–RO4*

He also stated that the patient should be aware of potential side effects. This awareness can help in patient compliance and fear reduction, which influences the patient's willingness to continue treatment. According to the participant, if proper counseling is provided, patients cooperate better with physicians and are better equipped to manage their symptoms.

*"One thing which the patient needs to know is basically the acute side effects, which may happen during the course of treatment because they should not be caught off guard, otherwise they'll not comply with the treatment. Let's say probably around 20 fractions is when the acute reaction usually starts setting in. So, at that particular point of time, if the patient's not already been primed for that, there's a chance that they might want to default from the treatment or they might be a little apprehensive about what is happening. So, we try to tell them that it's not easy treatment. There will be some difficulties along the way and we kind of prime them for that so that the patient also knows that probably at the 20 fractions mark or the four-week mark, there might be some acute reactions which they might encounter. And they also kind of co-operate regarding the do's and don'ts of whatever you say, they'll be a little bit more careful about that."*

*- RO4*

When asked about counseling regarding ototoxicity, one MO stated that ototoxicity doesn't receive significant emphasis. He stated that ototoxicity often impacts only the higher frequencies, and only when the cumulative dosage surpasses the thresholds will it influence the lower frequencies. Therefore, he did not perceive the necessity to emphasize ototoxicity during the counseling.

"No, actually we are explaining all the other toxicities. We are not stressing on the ototoxicity because usually it happens at the higher-frequency level. So as cumulative dose goes up, then only they'll have this hearing loss actually in the lower frequency levels."

–MO3

**Side effects of treatment of HNC and its management.**   The interview participants reported that RT and CRT can result in several side effects. The CT-associated side effects include renal toxicity, chemotherapy-induced nausea and vomiting (CINV), diarrhea, low blood count, nutritional deficiencies, and hyponatremia.

*"Renal toxicity, hyponatremia, hair loss, diarrhoea, chemo induced vomiting and the diarrhoea will be there for a few patients. . .. some of them they can have very low counts, low total counts, low platelet counts. They can have bleeding tendencies; they can have infections and land up with hypotension and shock in ICU. So, all these complications, whatever are there, we list out them and we tell them."*

*–MO2*

Radiation oncologists reported that patients often complained of xerostomia, mucositis, dysphagia, odynophagia, dental problems, and osteo-radio-necrosis. They stated that RT compromised the physiology of the structures on the affected side, which were usually the sites of irradiation. Since midline structures would be treated with wider fields of irradiation, bilateral structures would be affected. These side effects were reported to be acute and reversible. Nutrition was reported to be a crucial factor for effective treatment outcomes, and ROs prioritize maintaining optimal food intake throughout the treatment.

*"Yeah, the main issue, what they face during (treatment) is the acute xerostomia wherein the saliva—the patient's (saliva) consistency will vary and definitely there'll be some amount of dysphagia, because of that the radiation mucositis itself will be really painful. So, we tell them that—probably if it's well lateralized lesion, like a cheek lesions, then probably it's only gonna be an ipsilateral field which we are treating. So, the contralateral cheek will not have that much of a brunt of reactions. So, they might be able to kind of use the opposite side to kind of chew the food and also swallow. But if it's gonna be a midline structure like a tongue wherein definitely when you are treating bilateral neck as well, then the reactions are expected to be slightly more flawed. In such patients, we just prime them that probably a tube feed might be necessary because most of the practices still they follow insertion of Ryle's tube, SOS as and when it is required. But here in our institute, what we are trying to counsel the patient is for a prophylactic tube placement because nutrition plays a very important role. . .. So, what we try to prime them is that the nutrition plays an important role and if you're gonna be malnourished, the medical oncologist will definitely tell that because of nutrition grounds, we are not gonna go ahead with the fourth and fifth cycle of chemo. . .. We always keep them in touch with the nutritionist from day one. And that aspect is also reinforced. . .. If your feeding is secured, then the entire process becomes easy. Because more the acute reactions, like xerostomia or the mucositis, automatically their intake will decrease because the patient knows if*

*they're gonna consume the food, there'll be pain associated with that and there's a fear, which basically persists."*

*–RO4*

*"Dental problems we are seeing post radiation, post CT. So, proper dental evaluation, dental rehabilitation is going to be very, very important for all HNC patients who are going to get CT and radiation and who are going to be the long-term survivors. Definitely pre-op, post-op care cases also. Post-op, we are doing implantation and all. Generally, the old generation machines, they'd absorb more energy onto the bones because of the fixed energy levels. So, the bone toxicity, it is called osteo radio-necrosis (ORN)."*

*–RO5*

*"RO2: First two weeks usually goes on very smoothly. From the third week starts the grade one morbidity, mostly in the form of mucositis, pharyngitis, loss of taste, burning sensation, difficulty swallowing. It usually starts from the third weeks, peaks by about the fourth or fifth week. And it comes down maybe six weeks after radiation—completion of radiation, the symptoms start coming down and by three months they're back to normal.*

*Interviewer: So, it is acute and it is reversible?*

*RO2: Yes, mostly. Mostly, unless there is residual disease. In which case at the first follow up we would investigate them further and go in for a different modality of treatment, maybe surgery or maybe palliative CT or overall metronomic CT. As the situation vary; it varies with the age and the clinical. . .(scenario)."*

*–RO2*

Medical oncologists and radiation oncologists indicated that management of side effects involves the use of a grading scale to assess the severity through regular monitoring. If the side effects are classified as grade three or above, treatment is paused and symptomatic management is implemented.

*"If it is a three-week regimen, we have a weekly schedule of checking the total counts and see if he requires any other supportive measures for mucositis like—though the initially, um, if it's grade one, grade two, they generally take care–the radiation oncologist. And if it's more than grade two, then the patient is requiring some support system for the feed and all those things. Like we try to avoid CT. So, at that point of time, we will anyhow intervene and try to hold (pause) the treatment and then just treat in a symptomatic manner."*

*–MO2*

*"Grade three, grade four reactions will have very bad impact onto the nutritional intake. So, most of the patients who have grade three, grade four mucositis or side effects, these type of side effects, they may not be able to take appropriate nutrition. So, many a time we give a small break in the treatment or sometimes patient may need a nasogastric tube—through which patient may be given some sort of feeding. So, in case of a nutritional impact, we will stop the treatment."*

*–RO5*

None of the interview participants reported the grading of ototoxicity. Symptomatic management of the side effects includes prescribing a combination of medications for CINV,

topical applications for the management of mucositis, mouthwashes to prevent oral infections, analgesics for pain, etc.

*"The most common toxicity is nausea, vomiting with cisplatin. So, for that we have very good pre-medications like, three antagonists, and then NK1 receptors and a recent anti-psychotic. A combination of the all these drugs works."*

*–MO3*

*"For the mucositis, we give topical application of benzocaine and then we ask them to have frequent mouthwashes to prevent infection and also to prevent the infection aspect. And generally, analgesics for the throat pain. For the dermatitis, we give a solution called as Gentian violet."*

*–RO1*

For proactive management of these side effects, ROs stated that all patients with HNC undergoing RT are referred to dentists prior to treatment initiation. Patients are referred to speech and swallow therapists for rehabilitation of speech, swallow, and voice disorders secondary to treatment.

*"For HNC patients, before starting the radiation, usually we send them to the dentist because we are concerned with radio-necrosis of the mandible. Again, like the cochlea, this is another organ, which can be affected with radiation so we send them to the dentist and get the prophylaxis done for the oral cavity. . .. Speech, language and hearing sciences because, during radiation, the acute effects can be managed by exercises, like breathing exercise, swallowing exercise, and voice exercises, so that rehabilitation will be better."*

*–RO1*

Only one RO reported that patients with nasopharyngeal tumors are referred to an audiologist and an ophthalmologist due to the near proximity of the tumor site to the optic apparatus and cochlea. Baseline evaluation measures are completed, and follow-up measures may be performed to assess any changes.

*"Only in case of nasopharynx, since it's location itself is little critical in patients, what we routinely do is as a baseline, when we diagnose it's a CA nasopharynx, we'll ask for an audiometry and also an ophthalmologist's opinion as a baseline because there both the cochlea and the optic apparatus is very close. So, the dose will be a little higher than compared to a CA larynx or a hypo-pharynx. In those cases, as a baseline we take and post-treatment, maybe we'll again take up one more assessment just to see what is the difference."*

*–RO3*

### Theme 2—Ototoxicity

**Ear and hearing related side effects.**   The interview participants were probed regarding the ear and hearing-related side effects. One MO opined that the disease can itself cause changes in hearing, and cisplatin as a therapeutic agent can exacerbate these changes.

*"Some people who are having cisplatinum as a CT, there is a small chance of hearing loss. And some of the disease, per se, can cause defective hearing."*

*–MO1*

Another MO stated that hearing loss was a late toxicity and is usually reported by patients towards the end of treatment or during the follow-up.

*"With respect to CT, sometimes hearing loss is reported. But usually, the complain about this hearing will be at the end of treatment, maybe at the fifth cycle or sixth cycle, maybe during follow up months. It's not actually acute toxicity–it's almost delayed one. So, the complaint we usually get at the end of the treatment or during follow up."*

*–MO3*

Among ROs, there were conflicting reports of ear and hearing-related side effect. One RO reported that hearing loss was a late side effect.

*"Interviewer: Do you see any side effects related to hearing or hearing loss?*

*RO2: Uh, not in the acute phase. Not in the acute phase. The later stages they have."*

*–RO2*

Two other ROs reported that ear-related complaints are not late side effects and may be observed in the acute phase. However, these were reported to be temporary and included serous otitis media or aural fullness.

*"RO3: In cancers of the Nasopharynx, they don't present with hearing loss. They come in with some hearing complaints—ear discharge or some pain, like otitis media.*

*Interviewer: So, it was basically a temporary effect?*

*RO3: Yes, yes.*

*Interviewer: Any long-term effects on hearing or hearing loss?*

*RO3: Long term? I have never seen, no. No."*

*–RO3*

*"Rarely we see some patients reporting that I have a hard of hearing because of the radiation effect. Generally, they won't. In acute side effects, they'll say that I have some fullness because of the serous otitis media. Some sort of fluid collections may be there due to acute radiation effect. But generally, in the late toxicity profile, mostly they'll not tell us. Generally, the hard of hearing, because of the radiation effect or CT effect, nobody tells us."*

*–RO5*

When probed regarding the management of these ear and hearing-related complaints, the majority of the oncologists reported that these symptoms were treated symptomatically. Common management practices by ROs included obtaining an ENT opinion, and prescribing otic analgesics and antibiotics. These management strategies alleviated the symptoms.

*"Interviewer: How did you manage these complaints in these patients?*

*RO3: We used to get ENT opinion as we treat it like any normal patient. Like when some patient comes with an otitis media, how we treat with ear drops. That's it. That's the same way we treat them. ...So just basic drops and basic antibiotics. That's all. They'll be fine with it.*

*Interviewer: Okay. All the complaints got reversed?*

*RO3: Yeah. Yeah."*

*–RO3*

Only one MO emphasized the significance of baseline audiological evaluation as a management option. He believed that documenting the baseline hearing thresholds and monitoring the ototoxic effects was beneficial. He felt that this could influence the modification of the treatment and dose of cisplatin.

*"So usually, we should do a baseline audiometry to have a proper idea. Actually, see, some of the patients can have baseline hearing defect as well. So, to document that we should do baseline audiometry then follow up. Ideally it should be every CT patient so that we can pick up early ototoxicity and then we reduce the dose or we can completely stop the treatment."*

*–MO3*

**Perceived susceptibility to ototoxicity.**   The oncologist's perceptions regarding susceptibility to ototoxicity were explored. ROs believed that ototoxicity caused by RT was rare because only a limited dose of radiation reaches the cochlea; cochlear contouring is performed during the treatment planning phase. Hearing-related complaints were predominantly associated with the chemotherapeutic agent, which is ototoxic in nature, as opposed to RT.

*"Hearing I would say it is rare because in these three years I have seen a very few patients. Especially, to say this is a confirmed ototoxicity, I've seen only one patient. Otherwise, acute effects, definitely not with the hearing. Because the cochlea is also taken into account while planning for head and neck radiation. So, this cochlear dose will be within the limits when we give the radiation. It is the CT aspect that has a major effect on ototoxicity. So, when the patient is taking radiation along with CT weekly, definitely, they need to take into account the concern of ototoxicity, but acute effects, I have not seen a patient with hearing loss."*

*–RO1*

*"With radiation we mark out the cochlea. So, with IMRT we can also restrict the dose to the cochlea. So that is high on our priority if the tumor, what we call that PTV, PTV is the planning target volume. That's the final area that we are going to treat. If that is anywhere near the cochlea, we draw the cochlea and we make sure that we give the cochlea well below its tolerance, whenever we can. If the tumor is next to it or beyond, which is very close to the cochlea, then we can't save the cochlea then at least we save one side; that is the cochlea on one side. 35 Gy is the tolerance dose. Invariably we're below 35 Gy, unless in some nasopharyngeal cancers. If there's a parapharyngeal extension and we are not able to save, then we don't try to save (the cochlea). But most of the times we can save (the cochlea). In most HNCs except in nasopharynx where definitely the dose goes up to the limit or sometimes beyond."*

*–RO6*

One MO expressed the opinion that, given the patient's life-threatening illness and side effects, addressing hearing loss may not be the priority.

*"When HNC patient is suffering with a lot of problems and other things, his hearing may not be important for him. But I cannot say that it is not important for me as a quality-of-life issue. There are much more important concerns than this. . .. The hearing loss due to CT is not very profound. Even if you say cisplatinum, which is supposed to cause a hearing loss, it'll produce peripheral neuropathy, produce myelosuppression, it produces nephrotoxicity. Beyond that only the ototoxicity comes. . .. There are much more concerns with the patient . . .. Ultimately, it comes from (down) to the priority. If you have two things, you choose one which is more precious."*

*–MO1*

When inquired about the potential impact of an ototoxicity diagnosis on the physician's treatment decision, one RO reported that it would impact the choice of chemotherapeutic regimen. He stated that the treatment plan of the RO is not considerably affected by ototoxicity, as ROs already implement technical precautions to minimize the dose to the cochlea from the time of treatment initiation.

*"In case there is a reported ototoxicity, does that influence your treatment decision in any way?*

*RO6: No not mine, but definitely the medical oncologist. Because they would change from cisplatin to carboplatin or whatever. CT would change. So, if somebody is very little hearing in one ear, and I try my best to keep the dose as low as possible, but it won't influence the decision because with technology we can keep the doses low."*

*–RO6*

A RO stated that hearing loss seldom receives adequate consideration for two main reasons: oncologists may believe that the unaffected ear may compensate for the hearing loss. Alternatively, they may also believe that hearing loss can always be effectively managed with rehabilitative measures.

*"Probably (oncologists) feel that if it's unilateral they feel the contralateral ones can probably compensate or there are other kind of aids–hearing aids that are available, which can always be used and probably that is a light thing for them."*

*–RO4*

## Theme 3—Ototoxicity monitoring programs

The participants' awareness and perceptions of OMPs were explored during the interviews. This included questions regarding the current status of OMPs in their center and other centers across Tamil Nadu. All participants were aware of OMPs, possessed minimal knowledge of OMPs and understood its significance. However, none of them reported the implementation of an OMP in their center. They were unaware of any other centers implementing routine OMPs.

*"Ideally, just like how feeding is an important component which we enforce upon from day one, the ototoxic component also has to be evaluated on day one. Ideally all these patients should be submitted to a baseline audiometric testing and then one-month post-treatment*

*and then probably at the six months follow up it should be done. That is how the protocol basically defines it. But here, I don't see it being implemented in the Indian setting on a routine basis. . .. As far as I know from my friend's circle who are practitioners in other centres in Chennai, I don't think anyone has implemented. I mean it's almost the same everywhere and. . .. basically, I've done my training in Bangalore. So even Karnataka also, I don't think anything has followed this stringently. So, I think this is a pan India issue which has to probably be addressed."*

*–RO4*

All participants stated that there is scope for improvement in the evidence-base on ototoxicity in the country and felt that there is a need to investigate ototoxicity, particularly in rural settings.

*"There is scope for lot of improvement for the evidence base regarding ototoxicity. Abroad, yes, everything goes by evidence. In our country it is picking up including audiometry and other evidence-based practices. But by and large, uh, in the rural setups, it is still not picked up much. So, I think we should concentrate in that direction."*

*–RO2*

An RO admitted that he was unaware of the radiation tolerance dose and the contraindications for RT in patients with ear and hearing-related issues.

*"Evidence base is very less. . .. So, cisplatin, we know that if the creatinine clearance is less than 50, that's when it's contraindicated. Similarly, I don't know what is the Decibel level below which cisplatin would be contraindicated or what is the radiation tolerance dose, I don't know. So, we really don't have any data on that."*

*–RO6*

**Current practices of OM and challenges.** The interview participants were asked about their current practices of OM. An MO and RO stated that while they are aware of the importance of routine OM, they have never referred any patient to an ENT specialist or an audiologist for audiological evaluation. They reported that case history regarding hearing is not obtained frequently.

*"To be very frank, I have never referred anybody to the ENT surgeon or an audiologist to record what is the amount of hearing loss. Ideally before starting cisplatin, we have to do that. And then during the course we have to do the audiometry. But most of our patient, we ask them whether their hearing is normal. And this (hearing loss) we rarely come across."*

*–MO1*

*"Interviewer: Okay. And so how many patients do you actually get an audiogram done?*

*RO6: <Laugh>, maybe one in 75 or 100 actually. Sorry to say. . .. If there is baseline hearing loss, then I would monitor, but not beforehand, unfortunately not beforehand. So routinely we don't even ask any case history regarding hearing or hearing status."*

*–RO6*

Another MO stated that hearing loss will be investigated only if the patient complains of it. Hearing loss often goes undetected as patients commonly exhibit proficient communication and interaction with oncologists, causing oncologists to never notice any auditory changes. Furthermore, she believed that oncologists would be capable of monitoring for ototoxicity if they had previous clinical experience.

*"Hearing loss is something like–only if the patient tells me, I will think of it. Otherwise, we don't feel the need to ask because it's not going to be life threatening. We generally don't monitor so closely or vigilantly—we don't ask them every now and then. If they complain it's fine, if they don't complain, we just ask a question, are you fine? And that too, we don't even ask because if I'm seeing a patient, I ask him and he's responding to me. So, then I don't even ask him if he has a hearing loss, why would I ask it? So, we don't even address that. So, from our end also, we don't monitor it closely. So, if only I have some, I have done some work on this (ototoxicity), then I can ask them something."*

*–MO2*

Similarly, an RO reported that baseline audiological evaluations are only performed when hearing loss is suspected. This referral is subjective and is based on the oncologist's perceptions of the patient's hearing status during a conversation. Baseline audiograms were only used to ensure that the hearing loss was not caused by the treatment.

*"RO6: We get a baseline audiological evaluation.*

*Interviewer: Okay. For all of your HNC patients?*

*RO6: If they have a hearing loss, baseline hearing loss, then just to tell them that post radiation, it is not cause of the radiation. There's already a hearing loss beforehand.*

*Interviewer: So, on what basis do you decide if one patient needs audiogram versus another patient does not?*

*RO6: If there's a baseline hearing loss. We will know when we are speaking, they're not able to hear. Then then we will ask for baseline."*

*–RO6*

Only one RO reported performing baseline audiological evaluations on nasopharyngeal cancer patients who complained of ear and hearing problems. These patients were referred for an ENT opinion three months after their treatment, but follow-up audiological measures were not carried out. Routine OM is not performed for patients with other types of HNC.

*"Mostly for nasopharynx cases what we do is maybe after three months. Just to assess in case, they have any complaints, we do, we recheck. That's why we do our baseline because we want to make sure that our treatment has not affected them. . .. At the three-month follow-up, we just send opinion for ENT and ophthalmology. For nasopharynx advanced cases who has complaints. But otherwise for larynx, hypopharynx routinely we don't do because the dose doesn't go there."*

*–RO3*

According to the RO of a public charitable comprehensive cancer center, the medical oncology department uses a scale or a scoring system to screen for hearing loss and determine

the appropriate drug dose. Baseline audiograms are requested in cases of a pre-existing ear condition or hearing loss.

> "*Our medical oncology department screens them for auditory functions before giving CT. They have particular score. If it exceeds that, they go in for either carboplatin or a lower dose of cisplatin—weekly 40 milligram per square. So, which obviously does reduce that ototoxicity. . . .. . . . They have a scale. They have a score. If it's even one positive, they don't go in for that cisplatin or at least a hundred milligram per meter square. So that is only on that specific condition that in case there is. . .. Sometimes, they ask for an audiometry prior to giving CT. We don't ask for it but the medical oncology does ask for it in case there is some history of hearing loss or ENT problem, they ask for an audiometry. . .. In case there's a history of some hearing loss, partial hearing loss. Otherwise, they go by that scale.*"
>
> *–RO2*

However, the MO at the same centre indicated that only a history is recorded and baseline audiograms are not recommended for all patients.

> "*We don't do a baseline audiogram for all of our patients. We generally just only ask for history.*"
>
> *- MO2*

Another MO reported that patients are often monitored for vomiting, myelosuppression, tiredness, pain, etc., but not for ototoxicity. Baseline measures of echocardiogram and other tests are carried out; however, baseline audiometry is not performed for any patient due to logistic constraints.

> "*I don't ask any question regarding ototoxicity. I usually ask about the vomiting that is the most cumbersome thing, and then myelosuppression like fever, and then tiredness—those kind of things. And then any, pain that is related to disease as well as treatment, these are the main, areas of my concern. . . .. . .Actually, we are not doing baseline audiometry at all. In my practice, I'm not doing. We do baseline ECHO, ECG everything but, we don't practice baseline audiometry. Because of logistic issues.*"
>
> *–MO3*

The various challenges experienced by oncologists in undertaking OM were explored. According to one MO, some of these challenges include higher costs associated with audiological evaluation and increased testing duration.

> "*MO3: So, traditionally they have to go undergo some other tests. So, we charge maybe 2000 to 3000 and it will take quite some time as well. These are the reasons*
>
> *Interviewer: So, time and affordability of the major reasons.*
>
> *MO3: Yeah.*
>
> *Interviewer: So even if it takes some time for the patients to you know, complete the audiometry and come back, does it result in a delay in their treatment?*

*MO3: Actually, we are giving once in a week only. So, we can plan this two days before injection as well, but mainly it's the affordability."*

*–MO3*

Due to the lack of an in-house audiology department, the oncologist faced challenges in recommending patients for audiological evaluations. From the patient's perspective, undergoing audiological examinations was difficult due to a lack of financial and/or family support.

*"This centre being a charitable trust, we don't have an in-house audiology department and we have to send our patients outside. So not all of them will be able to afford it and even if they say affordability is not issue, but there's no support of people to take them there and human resources for the patient from the patient's side."*

*–MO2*

Another MO stated that patients may perceive a delay in their treatment due to the audiological evaluation. Audiological evaluation may not be a priority in comparison to the discomfort and pain they are feeling; they may not be healthy or comfortable enough to wait and undertake a comprehensive audiological evaluation.

*"They're (patients are) already feeling that the treatment is delayed or the diagnosis is delayed and will be wondering if it is going to take some more time because of this. I'm suffering from the pain—why worry about the hearing loss, which may or may not happen. . .. It is a question of being time consuming. You will make him sit among the 10 people and then you will see him. . .. . . he may be having some other problems which he may not be comfortable with sitting in the general population."*

*–MO1*

When patients were referred for audiological evaluation, there were ambiguous complaints of hearing loss by patients due to their lack of understanding. In addition, inaccuracies and ambiguity in the diagnosis of hearing loss in the audiological reports were a challenge.

*"There's some set of people where they say, like if you ask do you have hearing loss? They say yes. If you ask, you don't have any problem, right? Then they say yes, like depending on their understanding. Okay? But when we send them for audiogram, just because they're saying yes, they say mild hearing loss or sometimes the report is not proper. They just say mild hearing loss. They don't even mention whether it's a sensorineural or whether it's a conductive also. Sometimes they do give, but sometimes like the various places, so they don't give so."*

*–MO2*

An RO also reported that patients do not complain of hearing loss after treatment due to a lack of awareness or ignorance. Other symptoms and concerns take precedence over hearing loss, which may explain why oncologists do not monitor hearing on a frequent basis.

*"Even after completing the radiation, nobody tells us that they have developed some sort of hard of hearing from patient side to create some sort of awareness among the doctors. . .. ototoxicity due to radiation or CT from patient's side, at least, we are aware that we are reading in the books and that can cause ototoxicity. But from patient side, they're coming for the*

*dental problems coming forward and the complaining to us saying that there are damages in the teeth, honestly, nobody has told me so, far. They have never asked me, like after the treatment, I'm not hearing properly why, what is the problem? Never. . .. Patients are not telling to us. That's an important concern. If the patient has a nutritional problem, weight loss, pain, definitely they'll tell. This, I don't know whether they're bothered about it or not. We don't know. Patient may have, still may have, but they're not coming forward to tell us."*

*–RO5*

**Perceived barriers.** Oncologists' perspectives on the barriers to OMP implementation in the future were explored. According to one RO, stakeholders may exhibit reluctance to accept this program in its early phases, but this challenge can be addressed.

*"Initially, to implement any change, definitely there will be initial kind of hesitancies from all aspects, but that is part and parcel of any studies which we come across. That can always be overcome. I think once we are able to show exactly that a large number of patients are benefiting from all of this, then I'm sure people will definitely come around."*

*–RO4*

In addition, another RO expressed the difficulty of persuading patients to undergo regular audiological assessments, as they tend to prioritize initiating their cancer treatment promptly. He felt that it would be a challenge to convince them to undergo multidisciplinary consultations with dentists, ophthalmologists, and other specialists.

*"The negative aspect is that it should be accepted from all points. Like patient is already distressed and he is (presenting) with so many symptoms and to make them meet so many people (specialists) before the treatment. They all come with the mind-set that treatment is going to be done today and I'm going to get better tomorrow but oncology is not like that. It's a huge emotional form of treatment where emotions play a huge role, and it takes a lot of time for us to complete. We already ask them to meet the dentist separately, SLHS separately. This (Audiology) also comes in so—it should be acceptable from their (patient's) point of view. So that is one factor which will be challenging for us—to convince the patients."*

*–RO1*

Scheduling a mutually convenient time for audiological evaluations was also reported to be a challenge.

*"Our timings and the timings of Audiometry Department, that is the one issue."*

*–MO3*

The organization may perceive the establishment of an audiology clinic and the procurement of equipment as a costly undertaking. The OMP has to be cost-effective to ensure organizational acceptance and sustainability.

*"Cost factor of investment may be there; incorporating that department into an oncology hospital—the management has to agree, it should be cost effective for them. These may be the challenging factors."*

*–RO2*

Finally, one RO stated that he did not anticipate any major challenges during the implementation phase. He felt that challenges and solutions can only be identified when implementation begins.

*"I don't think there will be any challenges that cannot be worked out. It can be worked out. So, I don't think there should be any major issues. . .So almost, most of the things are available, it's only that we have to start using it and if you start using it, you'll know what are the pitfalls which are there and then probably you can fine tune things and work on it."*

*–RO4*

**Cues to action.**    All interviewees were asked for their suggestions and strategies to implement the OMP. One RO reported that an OMP can be implemented by utilizing existing frameworks and evidence found in the literature. He believed that this would allow for the prediction of the cost-benefit analysis. He also recommended collaborating with the institute's department of speech, language, and hearing sciences.

*"If we are able to select some parent articles, definitely it'll be enough for us to at least implement this at our institute level. . .. If we talk to speech language and hearing sciences—the instruments that they use and the cost—all this can be taken into account."*

*–RO1*

Interview participants were asked about the organizational support required for OMP implementation. One RO felt that adequate infrastructure and a dedicated audiology unit in the oncology department would be beneficial for counseling and the referral process. Similarly, another MO felt that it would be convenient to hire an audiologist dedicated to OM. This would eliminate the need for travel and ensure accessibility.

*"Maybe the management can give you (the audiologist) a good OP here (in radiation oncology department), actually, to sit and talk with the patient. Because, personally, if I am gonna meet a doctor, I feel more comfortable talking to them in a private space instead of just talking in the corridor or something like that. So maybe that kind of infrastructure I think that is little lacking even for you also to talk to them (the patients) or to examine. So that way I think if administration can help, it'll be great."*

*–RO3*

*"It'll be always helpful for us if we have an audiologist in-house. Because sometimes we only get a reports of mild hearing loss, which doesn't make any sense at all to me. My patient cannot afford to go back again, he cannot pay again and he requires some form of support for them to travel, go back, come back. I can't make him roam around. If I have the same thing in house, I can get it done easily. Whatever doubts I have, I can clarify. Like what is the amount of loss that he has."*

*–MO2*

Alternatively, an MO and RO suggested arranging regular audiology visits in the oncology unit to perform routine OM. They believed that this system would promote follow-up and facilitate effective collaboration between the audiologist and the oncologist.

*"Every week, on a particular day, an audiologist has to be here. Only then even we can refer the patient since the patient cannot wait for hours together. If they know that—say every Monday at two o'clock, a particular audiologist is there, they'll just walk into your OPD, maybe they can give their reviews or complaints and they can get back."*

*–RO3*

*"You need to allot a time for audiometry. See, if the audiologist is coming here and sitting with me for a week, he/she will know how many are going to come in a week. Then you can fix one particular day, one particular time or every day for example, between 10 to 11 is for cancer patients. You call them, do your tests, and send them."*

*–MO1*

Another MO proposed aligning the audiology clinic's operating hours with those of the cancer clinics to streamline referrals.

*"It should be accessible and open from eight to four. That would be useful so that we can refer the patient at any point of time in the OP office."*

*–MO3*

When asked about the need for human resources during the implementation process, one RO from a private quaternary care center stated that they had sufficient manpower. However, he perceived that the entire referral process requires optimization.

*"Basically, us being a medical college, we have enough resources in our hands in terms of postgraduate, interns—we have the allied health specialties, so there are a lot of people. So, I just think, the way you streamline it probably matters more."*

*–RO4*

The interview participants were asked about the multi-disciplinary role-players required in an OMP. A RO stated that both the oncologist and the audiologist should be involved throughout the entire counseling process, with the oncologist acting as the primary point of contact for the patient. The oncologist should support and enable the audiologist's involvement to ensure the patient's ease. Two other ROs agreed that the multidisciplinary team should include ENT specialists, clinical psychologists, and pain and palliative care specialists.

*"First and fore-most it's the oncologist as well as the audiologist. Unless the oncologist primes them and tells the patient, the patient will not be aware and unless they (the patient) know the audiologist cannot come into the picture."*

*–RO4*

*"Many of the patients get depressed. See the final consequences of an ototoxicity or a failure of treatment is psychological depression, suicidal tendencies. So definitely a psychologist, clinical psychologist should be there. That's also an essential part of any oncology treatments."*

*–RO2*

*"I think even pain and palliative care team is also a part of oncology only. But I think most of the centres are still missing out that team. Because it's not only about treatment being curative. Once we cross the curative aspect, the pain and palliative team, play a very vital role in*

*designing the treatment. Even during treatment also, because pain affects their quality of life. So, I think if a pain palliative and a clinical psychologist are also included, I think it it'll be complete for us."*

*–RO3*

According to a MO, the audiologist's primary responsibility would be to plan the management and rehabilitation of hearing loss.

*"The audiologist should be able to provide some help with the hearing aids equipment."*

*–MO3*

The oncologists were asked about strategies to increase patient acceptance of an OMP. An experienced RO suggested that patients might find OMPs to be useful if they were cost-effective and had a noticeable impact on their daily lives. It should be accessible and affordable.

*"For Indians, when it is very cheaper and it's going to be useful. So, the potential and impact onto the day-to-day practice is very, very important."*

*–RO5*

*"It (audiometry) should be easily accessible. It has to be easily accessible and it has to be in reach for the patient. So, if it is going to be some so sophisticated technique, if my patient is not going to afford it, if it's not going to help a large set of people, I don't think it is of much benefit."*

*–MO2*

An RO gave an interesting suggestion to consolidate the billing of allied health care services into a single package; the patient will not feel overburdened by the number of diagnostic evaluations they have to undergo and will believe that he is receiving valuable, multi-disciplinary care for his disease.

*"I don't think audiometry will cost that much. I don't think it'll be too expensive or something which cannot be borne by the patient. But what I will suggest is just like how we are charging the patients for radiotherapy treatments per se we can have a package which includes all the key components where the nutritionist exam, the baseline audiological testing and a clinical psychology exam, you can tell them (the patient) that all these things come as a part of a package. But if you tell them only audiology, they may not be willing, but if you're able to include, let's say as a part of your package and you tell them that this cost has already been included in the treatment cost, then the patient will not object. But if you're gonna quote it as a separate factor individually or separately they might say no or they might not comply."*

*–RO4*

Another RO believed that patients would quickly accept the new program provided they were adequately counseled and understood that routine audiological evaluations were being conducted for their own benefit.

*"They are the main players here. . .. As far as they're concerned, it's about them. So, it's about their recovery post-treatment. So, if we explain to them in a proper way, I think it would be*

*acceptable and the patient should be informed about everything that is being done because it's for their benefit at the end."*

*–RO1*

Likewise, the program's acceptance can be enhanced by prioritizing ototoxicity from the beginning and ensuring that the patient feels empowered in the decision-making process.

*"As a part of the initial discussion, when we are getting the consent, the ototoxicity part is not stressed at all. It's only the other things which we focus upon. If this is explained, I'm sure the patient will also be keen to know what options are available and what they need to do. Because definitely they'll not say that okay fine my hearing is gonna go and they'll be okay with it. They'll definitely want (to be intervened), so then obviously the discussion will grow and it is like a mutual give and take—the patient will also get to know and as an oncologist also, we'll know what are the facilities that are available. So that has to be conveyed to the patient in the first place."*

*–RO4*

Such a patient-centred approach will not only increase awareness of hearing loss but will also encourage patients to report any complaints of hearing loss to their team of specialists as soon as they begin experiencing challenges.

*"You should drive the patient to come forward to report any complaints of hearing loss. . .. They may not be aware also, sometimes they'll not tell us. They'll not tell to their relatives because noticing (hearing loss) itself is a big thing. Other people will only mention that he/she has hard of hearing. The impact of the symptom should drive the patient to come forward to tell to the treating physician. That is an important thing."*

*–RO5*

The interview participants were asked about the ideal timeline for conducting audiological evaluations. Both MOs and ROs agreed that audiological evaluations should be conducted before the initiation of cancer treatment and periodically throughout the treatment process. Subsequent assessments should be conducted at three months and six months post-treatment to account for late treatment effects.

*"These side-effects are immediate, not very delayed complications. So, say if I start with the first cycle, majority may not have (after) the first cycle, maybe a third—fourth cycle, the thresholds may start coming down, but the effect will there for at least six months' time. So, four cycles of chemo after that, during one follow up for three months, six months, then you can declare. At least three times you have to do for the same patient. First well before starting, midway during the treatment, at the end of treatment and after six months."*

*–MO1*

*"Usually, with respect to acute toxicities, we see up to three months. So during treatment and up to three months, it'll be only acute and sub-acute (in case there is a toxicity). So, if you're gonna stop within three months, you're gonna cover only till sub-acute. So, after three months*

*only the late toxicity sets it. So maybe at six months and then maybe yearly once if you want a long term follow up."*

*–RO3*

Regarding patient counseling, a RO opined that the responsibility of ototoxicity counseling should be initially assumed by the oncologist and thereafter by the audiologist. This would eliminate confusion and miscommunication between the team members. Engaging the services of a qualified psychologist during the treatment process can provide patients with a sense of support and assistance during their treatment.

*"The primary role of counselling should be by the oncologist and the secondary role can be that of an audiologist. Also, sometimes the patient can become very negative and apprehensive when they develop some appreciable deafness. So definitely, oncologist is a must and audiologist is also a must because they are the ones who can give hope to the patient saying that we have equipment to deal with it."*

*–RO2*

*"I think part of counselling has to be done by the oncologist and also the clinical psychologist also should be aware of such things because the first person the patient goes to is the clinical psychologist—because they do a weekly review for all our patients. . ...so I think both the oncologist as well as the clinical psychologist has to be aware of. . .If the primary counselling has already been done before treatment and then in subsequent visits, if the audiologist is gonna do, I don't think the oncologist also should be afraid of anything or there's no need for them to hesitate—then definitely the audiologist can do. But if the audiologist is the person who's gonna break the news to the patient in the first time, then definitely the blame game starts."*

*–RO4*

**Perceived utility.** Oncologists' perceptions regarding the possible impact of the OMP, it's potential success and their confidence to take action were explored during the SSIs. ROs believed that OMPs would provide reliable information regarding the patient's hearing status, aid in treatment planning, and also improve the overall quality of service delivery.

*"It (OMP) would definitely give us a proper result. If we identify ototoxicity, then we also have an idea on how to figure out the treatment for these patients. . .. If everything is in place, we would definitely get a proper result on who are getting affected, what is the main risk factor of getting affected and maybe we can formulate a treatment plan for them from that."*

*–RO1*

*"Definitely it (OMP) will improve quality (of service delivery). Especially when we are giving cisplatin and RT together."*

*–RO6*

Routine audiological evaluation and documentation of the hearing status could assist oncologists in reporting that the patient's hearing loss occurred prior to the treatment and was not caused by the treatment itself. This can help avoid any medico-legal issues in the future.

*"From our side, at least for the next patients, other patients we can try to do something in planning, like we can avoid it by doing IMRT. . . . . . That is a welcome thing—that is a good thing. . .. That is a face saver for the doctors also. Sometimes patients may not blame the treating physician upfront. . .. if we have the evidence of say pre-treatment damage, we can save our face. So, always this is going to be welcome sign and it may have impact on choosing the radiation technique, choosing the drug. . .. That should be the important point. . .. And if it is pre-treatment hearing loss, we can document it that it's not related to toxicity. So, medico-legally also, it may be better. Because baseline you have some evidence."*

*–RO6*

A MO stated that OM will allow for identification of hearing loss in pediatric patients, which can significantly influence their overall development.

*"Hearing is important because it enables communication with others and your social behaviour, especially in children—it'll affect their cognition, overall development, language and social interaction as well. So, I think it should be done in a systematic way. It'll be beneficial especially in children."*

*–MO3*

Another RO noted that the OMP should be disseminated to all healthcare professionals involved in cancer treatment to maximize its effectiveness.

*"So, it has to be primed on the whole to the doctor community . . .. the importance of this and I'm sure it'll come out."*

*–RO4*

According to an RO, routine OM will instil confidence in the stakeholders to continue it in a sustained manner.

*"So first it should be more routine. This baseline evaluation and follow up evaluation. It'll be more routine so that we are familiar with interpreting reports ourselves, at least to some extent. And then there also should be more clarity on what is the guideline—when to not give this particular medicine or beyond what dose we should not treat with radiation. Like if I see a CT scan now, I know how to interpret most of the time—70, 80% of the time, I get about 70% of the findings right by immediately seeing. But I've hardly seen any (Audiology) reports. I don't know how to interpret and so I just go with what the report is. So, because as I started seeing more and more, I get more confidence, I start referring more."*

*–RO6*

In addition, all oncologists expressed their willingness to collaborate with an audiologist for the OMP implementation. Almost all oncologists responded positively when asked if they would be willing to change their practice to include an OMP because an audiologist would be in charge of it. A RO added that context-specific challenges will arise only during the implementation; he felt that the OMP should be introduced with a few patients and modifications should be made during the implementation.

*"I don't think so we have to modify anything because it's just an add-on because from our side... There's nothing from our side to change or reduce. Since it's always a teamwork, we are ready to welcome audiologists into our team. That's all we can do. And we'll refer the cases and maybe monthly once we all can sit for a review for you to give your inputs also. Like if you're gonna say, so and so person is having such a toxicity, maybe we will go back and see our plans, what we have done, any modifications required. So basically, it's a teamwork. Only during our process, we will come up with all the hurdles only then we'll know what to change later."*

*–RO3*

## Discussion

The current study used a qualitative approach to ascertain the awareness and perspectives of ototoxicity and OM among oncologists treating individuals with HNC. Their perceptions regarding the barriers and strategies for OM were also explored.

The study findings revealed that all oncologists are aware of the ototoxic effects caused by RT and CT. However, they did not understand the impact of ototoxicity. They reported that hearing loss is rarely observed, and a case history regarding the hearing status is also not asked frequently. All oncologists reported that perceivable ototoxicity may be a late toxicity that patients typically report towards the end of treatment or during follow-up. While it is true that some patients may only report hearing loss after treatment is finished, research suggests that ototoxicity can occur during treatment as well [10, 31].

The ROs participating in the current study had varying opinions regarding the severity of the ototoxicity. There was a prevailing belief that ear and hearing-related complaints due to RT were uncommon. They consistently reported that the severity of hearing loss was not substantial enough to warrant significant concern. However, Brook et al. (2020) reported that patients undergoing RT for HNC can develop hearing loss as a later side-effect, especially when receiving a cumulative dose of 60 Gy. The authors also reported that the dose of radiation is directly proportional to the severity of hearing loss [32]. The common perception among the oncologists was that hearing was not considered a priority when the patient was experiencing other life-threatening complaints. These findings were similar to the findings of Santucci et al. (2021) [33]; the authors reported that audiological rehabilitation may not always receive the same level of attention or prioritization when the patient is under routine surveillance for his oncological treatment and resulting sequalae.

In the current study, it was interesting to note that ototoxicity was not emphasized by ROs and MOs during the counseling process. This could be due to an assumption that only a limited dose of radiation reaches the cochlea, thereby minimizing potential damage [34, 35]. However, it is important to recognize that while the recommended mean dose to the cochlea is ≤35 Gy to minimize the risk of SNHL [34], even lower doses can affect the cochlea [34]. Despite advancements in modern techniques that enable precise tumor targeting, some radiation exposure to the cochlea is difficult to avoid. Moreover, the combined impact of cumulative radiation and concurrent CT can further exacerbate the risk of ototoxicity [36, 37].

Additionally, the ROs believed that the chemotherapeutic agent had a higher likelihood of causing ototoxicity compared to RT. They felt that that it was the responsibility of the MO to completely evaluate the potential for hearing damage. While CT was reported to significantly contribute to hearing loss even at lower radiation doses [5, 9, 38], numerous studies indicate that radiation-induced hearing loss is a common side effect of radiation in individuals with

HNC [32, 39, 40]. A systematic review by Theunissen et al. (2014) reported that the incidence of RT-induced SNHL ranged from 0% to 43% [9]. This highlights that the shared responsibility of both ROs and MOs in assessing and managing ototoxicity risk. Collaborating with audiologists, and openly communicating the risks of ototoxicity to patients can greatly enhance management strategies.

One RO in the current study mentioned another reason for not emphasizing ototoxicity during counselling that there is a prevailing belief that if one cochlea is affected during the treatment, the contralateral cochlea has the potential to compensate for the loss. While it's true that the brain can adapt to some degree in unilateral hearing loss, relying solely on contralateral compensation can results in challenges in binaural hearing [41]. Depending on tumor location and radiation field, there is a risk of both cochleae receiving some level of radiation exposure [38]. This could lead to bilateral hearing loss, negating any potential benefit from contralateral compensation [38]. Although hearing loss is often viewed as manageable through rehabilitation since it is non-fatal it is important to recognize that adequate hearing is critical for quality of life and must not be overlooked [42].

Another reason for ototoxicity to be overlooked is due to the pattern of ototoxic hearing loss which initially affects the higher frequencies. Lower frequencies get affected later when the cumulative dose exceeds the recommended limits [9]. This delay in symptom presentation can result in a minimal impact on speech understanding. Thus, the treating physicians may miss early signs of hearing loss [43].

Typically, patients also do not report any hearing difficulties, possibly because they are unaware that their hearing loss could be a result of their cancer treatment, especially if the ototoxic risks of cancer treatment were not disclosed to them [14, 44]. Similarly, a study from South-Africa reported that there is often minimal disclosure/communication of the ototoxic effects of cancer treatments. This lack of disclosure could be due to healthcare professionals being unaware of these risks, or language barriers that affects clear communication between the healthcare professionals and patients [44].

Thus, it is imperative to enhance the knowledge and understanding of the ototoxic nature of cancer treatment modalities among oncologists. There is a significant need to update oncologists regarding the incidence and prevalence of ototoxic hearing loss induced by RT and CT. This finding aligns with previous studies [14, 31]; there is a need to increase awareness among patients and healthcare providers regarding the ototoxic nature of chemotherapeutic drugs. This would facilitate counseling and empower patients to promptly inform their treating physicians of any changes in their hearing status [14, 31]. Patients need to be informed of the early symptoms of ototoxicity to ensure its early identification and management [14, 31].

All participants in the current study demonstrated awareness and knowledge of OMPs, and understood its importance. However, none of them reported having implemented an OMP in their facility nor were they aware of any other centers regularly implementing such programs. Although oncologists acknowledge the importance of OM, they rarely refer patients to an ENT specialist or an audiologist. The referrals tend to be subjective and are based on the oncologist's perceptions of the patient's hearing status when communicating with them. These findings align with previous studies [14, 15, 44]. Garinis et al. (2018) reported the use of varied referral systems, ranging from routine referrals to audiology, and relying on patient self-referral. Similarly, Paken et al. (2020) found that while most oncologists reported referring the patient to the audiologist, the actual referrals were often made by nurses to ENT specialist. Moreover, patients were typically referred to audiologists, only when they reported having hearing difficulties, leading to a lack of baseline assessments. Thus, when compared to the theoretical awareness and knowledge of ototoxicity, the clinical translation is minimal [11]. Thus, it is crucial to raise awareness regarding the importance of OMPs and baseline audiological

evaluations among all physicians involved in cancer treatment [45]. Inputs from oncologists and other stakeholders is crucial to enhance monitoring schedules and improve coordination of OMP care with audiology services. Implementing of an electronic system for automatic referral generation for patients at risk for ototoxicity could facilitate early identification and routine monitoring [13, 14].

Both oncologists and audiologists need to be involved in the entire counseling process, with the oncologist serving as the primary contact for the patient [45]. The oncologist should facilitate the audiologist's involvement for the patient's ease. Prioritizing ototoxicity right from the beginning and ensuring that the patient feels autonomous in the decision-making process will facilitate treatment acceptance. Implementing a patient-centric approach will not only raise awareness of hearing loss, but also motivate patients to proactively communicate any hearing-related concerns to their team of specialists [14].

The SSIs explored the various reasons for the dearth of OM practices. The challenges include increased cost associated with audiological evaluation, long testing times, patients lacking the physical well-being or comfort to endure the wait for their audiological evaluation, ambiguous complaints of hearing loss by patients due to their lack of understanding, and errors and ambiguity in the diagnosis of hearing loss in the audiological reports. Lack of an in-house audiology department for ease of referrals was reported to be a significant challenge. From the patient's end, lack of affordability and/or family support were challenges to undergoing audiological evaluations. The major barriers anticipated by the oncologist in the future implementation of an OMP included (a) the initial cost to set up, (b) initial hesitation from stakeholders, (c) conflict in scheduling the audiological evaluation and the cancer treatment, and (d) difficulty convincing the patients of their hearing loss. The management must recognize that OMPs have a favourable cost-to-benefit ratio, as their approval is essential for the long-term sustainability of the OMP. These findings were similar to the barriers reported in other developed nations like the United States of America [46, 47]. Thus, the barriers to the implementation of OMPs in high-income countries (HICs) and LMICs are similar. The models implemented in HICs can be adapted for use in LMICs with suitable restructuring.

All interview participants were asked for their suggestions and strategies to sensitize and implement the OMP in a seamless manner. These suggestions included (a) adopting frameworks and evidence from existing literature for implementation; (b) establishing infrastructure to effectively address patient needs and provide counseling; (c) setting up protocols to streamline the referral process; and (d) having an in-house audiologist or audiology department to facilitate the interpretation of audiological reports and a seamless referral process. One interesting suggestion was to establish a regular schedule for an audiologist to visit the oncology ward for routine OM. All essential services for the treatment of HNC, including allied health-care, can be invoiced as a package. This may ensure that the patient does not feel overwhelmed by the associated cost of the extensive evaluation and instead feels he is receiving valuable, multi-disciplinary care for his condition. Konrad-Martin et al. (2018) proposed the utilization of screening methods to facilitate monitoring and incorporating audiological management into standard care [46]. The authors stated that adherence to guidelines might potentially be enhanced with the formal support of other stakeholders involved with OMPs. In addition, tele-audiology has proven to be an effective method for delivering OM [48].

The current study's findings regarding oncologists' confidence to take action and their perceptions about the potential efficacy of OMPs were unique. Radiation oncologists believed that OMPs would provide accurate results regarding the patient's hearing status, thereby facilitating better treatment planning and improved quality of service delivery. Additionally, they felt that it would facilitate documentation of patient's pre-existing hearing loss to mitigate potential medico-legal issues in the future. This highlights the importance of documentation in

ototoxicity monitoring; the findings of the audiological evaluation need to be documented rigorously with uniform reporting standards [13]. The findings can be integrated into an electronic database and automatic referral generation system [13, 14]. Similarly, a study from Italy reported that spontaneous reporting databases is a valid tool to be used in OM [49]. In the current study, all oncologists expressed their willingness to include an audiologist in the formulation of an OMP.

The results of the current study align with the key components of the health belief model [20], including perceived susceptibility, perceived barriers, and cues to action. By also examining current practices and factors related to ototoxicity, the study provides valuable insights into the status of OM in the South-Indian context. These perspectives and insights can be instrumental in developing a robust and comprehensive OMP. Understanding the barriers from the oncologists' viewpoints allows audiologists to address practical challenges related to logistics, resources, infrastructure, etc. [47]. Involving oncologists in the planning phase of the OMP can help tailor future research questions and solutions to improve quality of life, foster collaboration among healthcare professionals and produce actionable outcomes that may influence policy [47, 50].

Future directions of the current study include developing a comprehensive OMP by involving oncologists and ENT specialists as key stakeholders in cancer care centers. The findings of the current study can be used to structure a training program on ototoxicity and OM to increase awareness and knowledge to oncologists and audiologists [50]. Barriers identified from the current study could be addressed by using a portable, bedside equipment [51, 52]. Researchers can gain insights from patients after sustained implementation of OMPs to fine-tune the protocol.

The strength of the current study lies in its robust methodology. This study is among the first in South India to determine the perspectives, awareness, and knowledge of ototoxicity among oncologists treating individuals with HNC using a qualitative approach.

## Limitations

The current study is limited to findings from Chennai, a metropolitan city with well-established cancer care facilities, infrastructure and resources. The findings may be skewed to this geographical area and may not represent the status of OM in rural and sub-urban areas. In addition to this, incorporation of data triangulation could have increased the robustness of the study's findings. Despite efforts to increase the sample size of the medical oncology group, there were challenges in obtaining consent. This could affect the representativeness and generalization of the findings.

## Conclusion

This study explored the oncologist's awareness and perspectives on ototoxicity and OM for individuals with HNC in a South-Indian district through qualitative semi-structured interviews. The findings reveal that all oncologists are well aware of the ototoxic effects of RT and CRT, although these effects are not always emphasized during patient counselling. Despite the theoretical knowledge about OMPs, none of the oncologists reported implementing OMP programs in their facilities. Therefore, it is essential to enhance the knowledge and understanding of the ototoxic nature of cancer treatment modalities among oncologists. It is crucial to raise awareness regarding the significance of OMPs among all physicians involved in the treatment of patients with cancer. The OMP models implemented in HICs can be adapted for use in LMICs with suitable adaptation.

## Supporting information

**S1 Appendix. COnsolidated criteria for REporting Qualitative research (COREQ) checklist.**
(PDF)

**S2 Appendix. Interview guide for the semi-structured interviews.**
(DOCX)

## Acknowledgments

We would like to thank all the oncologists for their participation in the SSIs.

## Author Contributions

**Conceptualization:** Jayashree Seethapathy.

**Data curation:** Varsha Shankar.

**Formal analysis:** Varsha Shankar.

**Investigation:** Varsha Shankar.

**Methodology:** Varsha Shankar, Jayashree Seethapathy.

**Project administration:** Jayashree Seethapathy, Prasanna Kumar Saravanam.

**Resources:** Satish Srinivas.

**Supervision:** Jayashree Seethapathy, Prasanna Kumar Saravanam.

**Validation:** Jayashree Seethapathy.

**Visualization:** Varsha Shankar.

**Writing – original draft:** Varsha Shankar, Jayashree Seethapathy, Satish Srinivas, Raghu Nandhan, Prasanna Kumar Saravanam.

**Writing – review & editing:** Varsha Shankar, Jayashree Seethapathy, Satish Srinivas, Raghu Nandhan, Prasanna Kumar Saravanam.

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
