## [Decision Letter · Decision Letter 0]

7 Aug 2024

PONE-D-24-14073Current practices and perspectives of oncologists regarding ototoxicity monitoring in individuals with head and neck cancers undergoing radiation therapy and chemoradiotherapy: A qualitative study in a South Indian districtPLOS ONE

Dear Dr. Seethapathy,

Thank you for submitting your manuscript to PLOS ONE. After careful consideration, we feel that it has merit but does not fully meet PLOS ONE’s publication criteria as it currently stands. Therefore, we invite you to submit a revised version of the manuscript that addresses the points raised during the review process.

I would like to sincerely apologise for the delay you have incurred with your submission. It has been exceptionally difficult to secure reviewers to evaluate your study. We have now received two completed reviews; the comments are available below. The reviewers have raised significant scientific concerns about the study that need to be addressed in a revision.

Please revise the manuscript to address all the reviewer's comments in a point-by-point response in order to ensure it is meeting the journal's publication criteria. Please note that the revised manuscript will need to undergo further review, we thus cannot at this point anticipate the outcome of the evaluation process.

We look forward to receiving your revised manuscript.

Kind regards,

Miquel Vall-llosera Camps

Senior Staff Editor

PLOS ONE

Journal Requirements:

Reviewers' comments:

Reviewer's Responses to Questions

**Comments to the Author**

1. Is the manuscript technically sound, and do the data support the conclusions?

Reviewer #1: No

Reviewer #2: Partly

2. Has the statistical analysis been performed appropriately and rigorously? 

Reviewer #1: N/A

Reviewer #2: N/A

3. Have the authors made all data underlying the findings in their manuscript fully available?

Reviewer #1: Yes

Reviewer #2: Yes

4. Is the manuscript presented in an intelligible fashion and written in standard English?

Reviewer #1: Yes

Reviewer #2: Yes

5. Review Comments to the Author

Reviewer #1: The sectional headings should be arranged correctly. There critical information missing from the manuscript. The authors should describe the setting in enough detail to give the reader a clear picture. The research approach should be highlighted in order to allow the reader to identify the coherence in the methods used in the study. The authors should include a section to clearly describe all ethical considerations / issues pertaining to the study. Furthermore, the authors should clarify the inclusion and exclusion criteria instead of stating that they ensured certain qualities existed within the selected population.

The results section should be reorganized as the current form does not read well. All excerpts must be included in text to show how the quotations fit in the narrative element. I have shared an article in the track changes to guide and support this.

The discussion could benefit form more literature support as there is large pieces of paragraphs where the authors have just written form the heart.

The authors should create a separate heading for the limitations and elaborate on how these limitations impacted their study.

Include a conclusion to provide a summary of the whole study.

Reviewer #2: Current practices and perspectives of oncologists regarding ototoxicity monitoring in individuals with head and neck cancers undergoing radiation therapy and chemoradiotherapy: A qualitative study in a South Indian district

Overall comment

In a nutshell, this study explores oncologists' awareness and perspectives on ototoxicity monitoring for head and neck cancer patients in South India through qualitative interviews. The findings highlight the need to enhance oncologists' understanding of ototoxicity and advocate for monitoring programs. Given the significant impact on child development, academic achievement, balance, and mental health, this study's implications are critical. However, there are some suggestions to enhance the paper's quality.

Title

Consider shortening the title of your study to less than 15 words. A shorter title can better grab readers' attention and convey the main idea concisely. My suggestions could be;

1. “Oncologists' views on ototoxicity monitoring in head and neck cancer patients: A South Indian qualitative study”

2. “Exploring Ototoxicity Awareness Among Oncologists in South Indian Cancer Care”

Abstract

Methods – COREQ is an abbreviation, in this instance, it has been used without a prior definition. State in full before using it.

Keywords

Is “semi-structured” appropriate as a keyword?

Introduction

The introduction is well-written and effectively identifies the knowledge gap. However, a few points need to be addressed;

• Line 41 – grammatical error “In India, head and neck cancers (HNC) are one” not “is”

• Lines 45-47 – the sentence is too long and loses its meaning. Consider breaking it down into two or rephrasing it

• Line 51 – grammatical error “the” missing, treatment depends upon the stage

• Line 58 – Avoid starting a sentence with an abbreviation

Methods

• The COREQ guidelines mandate including a section that identifies the research design and provides the rationale for its selection.

• Line 111 – The abbreviation for ROs has already been defined, no need to repeat it here

• Interview details – Could you please provide a list of the interview questions that were included in the interview guide?

• Analysis – To provide clarity, it would be beneficial to specify the approach used in thematic analysis, such as employing the method outlined by Braun and Clarke. Thematic analysis is too broad

• Providing a citation and a brief explanation of what QDA Miner Lite is would be beneficial for the readers.

• While efforts have been made to describe measures for ensuring the trustworthiness of the study, it is recommended to have a dedicated section that explicitly outlines this process. Providing a detailed explanation of how your study, for instance, addresses all four criteria for assessing trustworthiness, as outlined by Lincoln and Guba, will enhance the credibility of the methods section.

Results

• I am curious about how to navigate this section. Will readers need to refer back to the quotes while simultaneously reading the results section? Why not follow the conventional approach of integrating the quotes within the results section for readability?

• Lines 221 and 227 – Avoid starting sentences with abbreviations. Correct this throughout the document

Discussion

• Lines 402-420, feel like a rehash of the findings as there is no attempt to link the stated findings to existing literature.

• Line 421 – “Contrary to beliefs of oncologists” Which oncologists? From this study or?

The statement further says numerous studies but only one study is cited. Please cite those studies

• Line 422 – include the names of the authors of the systematic review. Like “A systematic review by Theunissen et al., …”

• Line 426 – “This finding aligns with previous studies [14]” but only one study is cited, please cite the studies otherwise change to “This finding aligns with a previous study [14]”

• There is a need for a discussion on how the study’s findings align or misalign with the theoretical framework. Furthermore, suggestions for further research on how the framework employed can be improved are needed.

• There is no attempt to explore potential avenues for future research in general

• Line 475-481 – there is no attempt to link the findings to existing literature

References

Ok

6. PLOS authors have the option to publish the peer review history of their article (what does this mean?). If published, this will include your full peer review and any attached files.

Reviewer #1: **Yes: **Dr L Gumede

Reviewer #2: No

---

## [Author Response · Author response to Decision Letter 0]

6 Sep 2024

From, 

Jayashree S

Associate Professor and Head

Department of Audiology 

Sri Ramachandra Faculty of Audiology and Speech, Language Pathology

SRIHER (DU), Chennai, India

To, 

The Editor

PLOS ONE

Dear Editor,

Sub: Response to reviewer’s comments on the manuscript - PONE-D-24-14073

I would like to thank you for considering our manuscript, “Current practices and perspectives of oncologists regarding ototoxicity monitoring in individuals with head and neck cancers undergoing radiation therapy and chemoradiotherapy: A qualitative study in a South Indian district” for publication in your esteemed journal. Upon the reviewer’s recommendation, we have changed the title to, “Oncologists' views on ototoxicity monitoring in head and neck cancer patients: A South Indian qualitative study.”

I take this opportunity to extend my gratitude to the reviewers for their valuable comments. Their suggestions and recommendations helped us improve our manuscript. We have answered all the reviewer’s comments in the document attached herewith. All recommendations have been incorporated in the manuscript and the changes are highlighted in yellow.

We look forward to hearing from you regarding the status of our submission.

Thanking you,

Sincerely,

Jayashree S 

Response to comments of Reviewer #1: Comments in the PDF file

1. Line 50 Surgical oncologist? Medical Oncologist? - Write in full 

Response: This change has been incorporated as recommended by the reviewer (Page 3, Line 50)

2. Line 58 Write in full at the beginning of the sentence

Response: This sentence has been rephrased for better grammar and readability (Page 3, Line 58)

3. Line 100-101 What research approach did the researcher employ? 

Response: The study used a qualitative approach and a cross-sectional snapshot study design was used. This has been added in the method (Page 6, Lines 100 – 104)

4. Line 99-100 include a section maybe after data analysis to discuss the ethical considerations or issues. 

Response: This change has been incorporated as recommended by the reviewer. A section on ethical considerations has been elaborated after the data analysis in the method section (Page 9, Lines 177 - 184)

5. Line 111 Where? not the actual centers but the area (Country/Province/State). 

Response: We thank the reviewer for this valuable comment. This change has been incorporated as recommended by the reviewer (Page 7, Line 125)

6. Line 112 How? This makes it seem as if the researcher provided some form of intervention to 'ensure' the participants were knowledgeable. I suggest working on the sentence construction here for clarity if this was part of the inclusion criteria 

Response: We thank the reviewer for this valuable comment. This sentence has been edited accordingly as recommended by the reviewer (Page 7, Lines 126 - 128)

7. Line 117 How were the participants contact details obtained? 

Response: The researchers obtained the contact numbers of several oncologists through known contacts and colleagues in the field of medical and radiation oncology. This information has been added as recommended by the reviewer (Page 7, Lines 132 - 133)

8. Under interview details, were there no field notes take during the on-site interviews? 

Response: The interviews were audio recorded and field notes were maintained during the interviews. This has been mentioned in the method (Page 8, Lines 148 - 149)

9. Line 135 What does QDA stand for? write in full, since this is the only place intext where the acronym is used 

Response: QDA stands for Qualitative Data Analysis. This change has been incorporated as recommended by the reviewer (Page 8, Lines 155 - 156)

10. Line 154 I suggest tabulating the information that follows. Also tabulate this data to clearly indicate if the bulleted points represent sub-themes or categories. 

Response: We thank the reviewer for this valuable comment. This change has been incorporated as recommended by the reviewer (Page 12, Line 205)

11. Results: I suggest including the quotes in text. The current format is not reader friendly and is very confusing. Suggested read: Cristancho, S., Watling, C.J. and Lingard, L.A. 2021. Three principles for writing an effective qualitative results section. Focus on Health Professionals, VOL. 22, NO. 3, 111- 124

Response: We thank the reviewer for this valuable comment. The quotes have been included in the results section (Pages 12 - 38)

12. Line 175 I suggest including quotes in the text 

Response: This change has been incorporated as recommended by the reviewer (Pages 12 - 38)

13. Line 191 I suggest including quotes in the text 

Response: This change has been incorporated as recommended by the reviewer (Pages 12 - 38)

14. Line 459 which countries are being referenced? 

Response: The authors referenced to the United States of America in this statement. The statement has been edited accordingly (Page 42, Line 979)

15. Line 145 discuss limitations separately and elaborate how the limitations impacted the study. 

Response: We thank the reviewer for this valuable comment. This change has been incorporated as recommended by the reviewer (Page 44, Lines 1023 - 1029)

Response to comments of Reviewer #1: 

16. The sectional headings should be arranged correctly. 

Response: This change has been incorporated as recommended by the reviewer. The sectional headings are arranged according to the COREQ checklist (Pages 6 - 10)

17. There critical information missing from the manuscript. The authors should describe the setting in enough detail to give the reader a clear picture. 

Response: This change has been incorporated in the methods section as recommended by the reviewer. In addition, table 1 provides details regarding the setting of the interview participants who consented to participate in the study (Page 8, Lines 141 - 143)

18. The research approach should be highlighted in order to allow the reader to identify the coherence in the methods used in the study. 

Response: The study incorporated a qualitative research approach. This has been mentioned in the methods section (Page 6, Lines 100 - 104)

19. The authors should include a section to clearly describe all ethical considerations / issues pertaining to the study. 

Response: This change has been incorporated in the methods section after data analysis as recommended by the reviewer (Page 9, Lines 177 - 184)

20. Furthermore, the authors should clarify the inclusion and exclusion criteria instead of stating that they ensured certain qualities existed within the selected population. 

Response: This change has been incorporated as recommended by the reviewer (Page 7, Lines 126 - 131)

21. The results section should be reorganized as the current form does not read well. All excerpts must be included in text to show how the quotations fit in the narrative element. I have shared an article in the track changes to guide and support this. 

Response: This change has been incorporated as recommended by the reviewer (Pages 12 - 38)

22. The discussion could benefit form more literature support as there is large pieces of paragraphs where the authors have just written form the heart. 

Response: This change has been incorporated as recommended by the reviewer. The discussion has been edited to include more information and literature support (Pages 38 - 44)

23. The authors should create a separate heading for the limitations and elaborate on how these limitations impacted their study. 

Response: This change has been incorporated as recommended by the reviewer (Page 44, Lines 1023 - 1029)

24. Include a conclusion to provide a summary of the whole study. 

Response: This change has been incorporated as recommended by the reviewer (Page 44, Lines 1030 - 1039) 

Response to comments of Reviewer #2:

25. Title: Consider shortening the title of your study to less than 15 words. A shorter title can better grab readers' attention and convey the main idea concisely. My suggestions could be;

1. “Oncologists' views on ototoxicity monitoring in head and neck cancer patients: A South Indian qualitative study”

2. “Exploring Ototoxicity Awareness Among Oncologists in South Indian Cancer Care” 

Response: This change has been incorporated as recommended by the reviewer. The current title is now, “Oncologists' views on ototoxicity monitoring in head and neck cancer patients: A South Indian qualitative study” (Page 1, Line 1)

26. Abstract: Methods – COREQ is an abbreviation, in this instance, it has been used without a prior definition. State in full before using it. 

Response: This change has been incorporated as recommended by the reviewer (Page 2, Line 23)

27. Keywords: Is “semi-structured” appropriate as a keyword? 

Response: “Semi-structured” has been removed from the keyword list (Editorial Manager Portal; Manuscript data; Keywords)

28. Line 41 – grammatical error “In India, head and neck cancers (HNC) are one” not “is” 

Response: This change has been incorporated as recommended by the reviewer (Page 3, Line 41)

29. Lines 45-47 – the sentence is too long and loses its meaning. Consider breaking it down into two or rephrasing it 

Response: This change has been incorporated as recommended by the reviewer (Page 3, Lines 45 - 47)

30. Line 51 – grammatical error “the” missing, treatment depends upon the stage 

Response: This change has been incorporated as recommended by the reviewer (Page 3, Line 51)

31. Line 58 – Avoid starting a sentence with an abbreviation 

Response: This change has been incorporated as recommended by the reviewer (Page 3, Line 58)

32. The COREQ guidelines mandate including a section that identifies the research design and provides the rationale for its selection. 

Response: This change has been incorporated as recommended by the reviewer (Page 6, Lines 100 - 104)

33. Line 111 – The abbreviation for ROs has already been defined, no need to repeat it here 

Response: This change has been incorporated as recommended by the reviewer (Page 7, Line 125)

34. Interview details – Could you please provide a list of the interview questions that were included in the interview guide? 

Response: The interview guide has been provided as figures 2 and 3, in addition to the appendix file (Page 8, Lines 145, 152, Figs 2 and 3, S2 Appendix 2)

35. Analysis – To provide clarity, it would be beneficial to specify the approach used in thematic analysis, such as employing the method outlined by Braun and Clarke. Thematic analysis is too broad 

Response: This change has been incorporated as recommended by the reviewer (Page 8, Lines 155 - 172)

36. Providing a citation and a brief explanation of what QDA Miner Lite is would be beneficial for the readers. 

Response: This change has been incorporated as recommended by the reviewer (Page 8, Lines 155 - 157)

37. While efforts have been made to describe measures for ensuring the trustworthiness of the study, it is recommended to have a dedicated section that explicitly outlines this process. Providing a detailed explanation of how your study, for instance, addresses all four criteria for assessing trustworthiness, as outlined by Lincoln and Guba, will enhance the credibility of the methods section. 

Response: We thank the reviewer for this interesting suggestion. This change has been incorporated as recommended by the reviewer. The measures to ensure trustworthiness has been described in the methods section (Page 10, Lines 185 - 198)

38. I am curious about how to navigate this section. Will readers need to refer back to the quotes while simultaneously reading the results section? Why not follow the conventional approach of integrating the quotes within the results section for readability? 

Response: This change has been incorporated as recommended by the reviewer. The quotes have been integrated within the results section for readability (Pages 12 - 38)

39. Lines 221 and 227 – Avoid starting sentences with abbreviations. Correct this throughout the document 

Response: This change has been incorporated as recommended by the reviewer (Page 17, Line 340)

40. Lines 402-420, feel like a rehash of the findings as there is no attempt to link the stated findings to existing literature. 

Response: This change has been incorporated as recommended by the reviewer. The discussion has been edited to include more information and literature support (Pages 39 - 40)

41. Line 421 – “Contrary to beliefs of oncologists” Which oncologists? From this study or?

The statement further says numerous studies but only one study is cited. Please cite those studies 

Response: This sentence has been edited for better flow and readability. The subsequent statement regarding numerous studies has been edited to include the citations of the respective studies (Page 39, Lines 912 - 916)

42. Line 422 – include the names of the authors of the systematic review. Like “A systematic review by Theunissen et al., …” 

Response: This change has been incorporated as recommended by the reviewer (Page 40, Line 916)

43. Line 426 – “This finding aligns with previous studies [14]” but only one study is cited, please cite the studies otherwise change to “This finding aligns with a previous study [14]” 

Response: This change has been incorporated as recommended by the reviewer (Page 41, Line 941)

44. There is a need for a discussion on how the study’s findings align or misalign with the theoretical framework. Furthermore, suggestions for further research on how the framework employed can be improved are needed. 

Response: This change has been incorporated in the discussion as recommended by the reviewer (Page 43, Lines 1005 - 1013)

45. There is no attempt to explore potential avenues for future research in general 

Response: This change has been incorporated as recommended by the reviewer. The future directions of the study have been elaborated in the discussion (Page 43, Lines 1014 - 1019)

46. Line 475-481 – there is no attempt to link the findings to existing literature 

Response: This change has been incorporated as recommended by the reviewer. The discussion has been edited to include more information and literature support (Pages 43, Lines 995 – 1004)

---

## [Decision Letter · Decision Letter 1]

4 Oct 2024

PONE-D-24-14073R1Oncologists' views on ototoxicity monitoring in head and neck cancer patients: A South Indian qualitative studyPLOS ONE

Dear Dr. Seethapathy,

Thank you for submitting your manuscript to PLOS ONE. After careful consideration, we feel that it has merit but does not fully meet PLOS ONE’s publication criteria as it currently stands. Therefore, we invite you to submit a revised version of the manuscript that addresses the points raised during the review process.

We look forward to receiving your revised manuscript.

Kind regards,

Ashish Wasudeo Khobragade, MD

Academic Editor

PLOS ONE

Journal Requirements:

**Additional Editor Comments:**

1. I recommend starting the abstract with a background or introduction and ending it with a conclusion instead of a discussion section.

2. The materials and methods section is started with ethical permission. This statement may be placed under the ethical consideration heading. There is no connectivity between the sentences in the first paragraph. Organize the materials and methods section to maintain connectivity between the sentences and paragraphs.

Reviewers' comments:

Reviewer's Responses to Questions

**Comments to the Author**

1. If the authors have adequately addressed your comments raised in a previous round of review and you feel that this manuscript is now acceptable for publication, you may indicate that here to bypass the “Comments to the Author” section, enter your conflict of interest statement in the “Confidential to Editor” section, and submit your "Accept" recommendation.

Reviewer #1: All comments have been addressed

Reviewer #2: All comments have been addressed

2. Is the manuscript technically sound, and do the data support the conclusions?

Reviewer #1: Yes

Reviewer #2: Yes

3. Has the statistical analysis been performed appropriately and rigorously? 

Reviewer #1: N/A

Reviewer #2: N/A

4. Have the authors made all data underlying the findings in their manuscript fully available?

Reviewer #1: Yes

Reviewer #2: Yes

5. Is the manuscript presented in an intelligible fashion and written in standard English?

Reviewer #1: Yes

Reviewer #2: Yes

6. Review Comments to the Author

Reviewer #1: I would like to thank the authors for their efforts and consideration of the comments and suggestions. I believe that the additional detail has improved this paper. The minor comments provided can be applied based on the authors' editor's decision.

Reviewer #2: (No Response)

7. PLOS authors have the option to publish the peer review history of their article (what does this mean?). If published, this will include your full peer review and any attached files.

Reviewer #1: **Yes: **Dr Lindiwe Gumede

Reviewer #2: No

---

## [Author Response · Author response to Decision Letter 1]

9 Oct 2024

Response to comments of Reviewer #1: Comments in the PDF file

Line 23 include the correct research design. COREQ is not a research design, it is a checklist for reporting of studies using interviews and focus groups. 

Response: This change has been incorporated as recommended by the reviewer (Page 2, Lines 25 - 26)

Line 103 Use this as the first sentence

Response: This change has been incorporated as recommended by the reviewer (Page 7, Lines 108 - 109)

Line 157 Remove (2006, 2012) 

Response: This change has been incorporated as recommended by the reviewer (Page 9, Line 163)

Response to comments of Editor: 

I recommend starting the abstract with a background or introduction and ending it with a conclusion instead of a discussion section. 

Response: This change has been incorporated as recommended by the editor. The sectional headings are arranged according to the COREQ checklist (Pages 2 - 3)

The materials and methods section is started with ethical permission. This statement may be placed under the ethical consideration heading. There is no connectivity between the sentences in the first paragraph. Organize the materials and methods section to maintain connectivity between the sentences and paragraphs. 

Response: This change has been incorporated in the methods section as recommended by the reviewer. 

The method of the study is presented according to sectional headings of the COREQ checklist such as, “Research team and reflexivity”, “Study design”, “Analysis and findings”. (Pages 7 - 11)

---

## [Editor Report · Decision Letter 2]

15 Oct 2024

Oncologists' views on ototoxicity monitoring in head and neck cancer patients: A South Indian qualitative study

PONE-D-24-14073R2

Dear Dr. Seethapathy,

We’re pleased to inform you that your manuscript has been judged scientifically suitable for publication and will be formally accepted for publication once it meets all outstanding technical requirements.

Kind regards,

Ashish Wasudeo Khobragade, MD

Academic Editor

PLOS ONE
---

## [Editor Report · Acceptance letter]

17 Oct 2024

PONE-D-24-14073R2 

PLOS ONE

Dear Dr. Seethapathy, 

I'm pleased to inform you that your manuscript has been deemed suitable for publication in PLOS ONE. Congratulations! Your manuscript is now being handed over to our production team.

Kind regards, 

on behalf of

Dr. Ashish Wasudeo Khobragade 

Academic Editor

PLOS ONE